# Impact of moderately energetic fine-scale dynamics on the phytoplankton community structure in the western Mediterranean Sea

Roxane Tzortzis[1], Andrea M. Doglioli[1], Stéphanie Barrillon[1], Anne A. Petrenko[1], Francesco d'Ovidio[2], Lloyd Izard[2], Melilotus Thyssen[1], Ananda Pascual[3], Bàrbara Barceló-Llull[3], Frédéric Cyr[4], Marc Tedetti[1], Nagib Bhairy[1], Pierre Garreau[5], Franck Dumas[6], and Gérald Gregori[1]

[1]Aix Marseille Univ., Université de Toulon, CNRS, IRD, MIO, Marseille, France
[2]Sorbonne Université, CNRS, IRD, MNHN, Laboratoire d'Océanographie et du Climat : Expérimentations et Approches Numériques (LOCEAN-IPSL), Paris, France
[3]IMEDEA (CSIC-UIB), Instituto Mediterráneo de Estudios Avanzados, Esporles, Spain
[4]Northwest Atlantic Fisheries Centre, Fisheries and Oceans Canada, St. John's, NL, Canada
[5]UMR 6523 CNRS, IFREMER, IRD, UBO, Laboratoire d'Océanographie Physique et Spatiale, Plouzané 29280, France
[6]SHOM, Service Hydrographique et Océanographique de la Marine, 13 rue de Chatellier, CS592803, 29228 Brest, CEDEX 2, France

**Correspondence:** Roxane TZORTZIS (roxane.tzortzis@mio.osupytheas.fr)

**Abstract.**

Model simulations and remote sensing observations show that ocean dynamics at fine scales (1–100 km in space, day–weeks in time) strongly influence the distribution of phytoplankton. However, only few in situ samplings have been performed and most of them in western boundary currents which may not be representative of less energetic regions. The PROTEVSMED-SWOT cruise took place in the moderately energetic waters of the western Mediterranean Sea, in the southern region of the Balearic Islands. Taking advantage of near-real time satellite information, a sampling strategy was defined in order to cross a frontal zone separating different water masses. Multi-parametric in situ sensors mounted on the vessel, on a towed vehicle and on an ocean glider were used to sample at high spatial resolution both physical and biogeochemical variables. Particular attention was put in adapting the sampling route, in order to also estimate the vertical velocities in the frontal area. Such a strategy was successful in sampling quasi-synoptically an oceanic area characterized by the presence of a narrow front with an associated vertical circulation. A multiparametric statistical analysis of the collected data identifies two water masses characterized by different abundances of several phytoplankton cytometric functional groups, as well as different contents in chlorophyll $a$ and $O_2$. With respect previous studies, here we focus on moderately energetic fronts induced by fine-scale circulation. Moreover, we explore physical-biological coupling in an oligotrophic region. Our results show that the fronts induced by the fine-scale circulation, even if weaker than the fronts occurring in energetic and nutrient-rich boundary current systems, maintain a strong structuring effect on phytoplankton community by segregating different groups at the surface. Since oligotrophic and moderately energetic regions are representative of a very large part of the world ocean, our results can be extrapolated and have global significance.

# 1 Introduction

Phytoplankton is essential for the functioning of ocean, by supporting the marine food chain and playing a key role in biogeochemical cycles. It is responsible for almost half of the oxygen produced each year on the planet by photosynthesis (Field et al., 1998). In particular, its role on $CO_2$ recycling (Watson et al., 1991) is important to study in the context of global climate change. Ptacnik et al. (2008) have also shown the importance of phytoplankton diversity for the functioning of the oceanic ecosystem. Since several decades, satellite observations have revealed that phytoplankton concentrations at the surface of the ocean are characterized by a patchy distribution (Gower et al., 1980; Yoder et al., 1987). These patches can be generated by biological processes such as cell buoyancy, behavioral patterns or grazing (Martin, 2003), but also by physical ocean circulation (Strass, 1992). The term "fine scales" refers here to ocean dynamical processes occurring on horizontal scales of the order of 1–100 km, characterized by a small Rossby number, and a relative short lifetime from days to weeks. Such ephemeral structures are induced by mesoscale interactions and frontogenesis (McWilliams, 2016). Since the lifetime of fine-scale is often similar to the phytoplankton growth time scale, this suggests that fine-scale can affect and modulate the phytoplankton community.

The role of fine scales on phytoplankton bulk primary production is now well established. Several fields and modelling studies have shown that fine-scales could generate intense vertical velocities, which transport nutrients from the mixed layer to the euphotic zone, enhancing the phytoplankton production (e.g., Lapeyre and Klein, 2006; Lévy et al., 2001; Pidcock et al., 2016; Mahadevan, 2016). These vertical motions can also limit the phytoplankton growth, subducting phytoplankton cells from the euphotic zone to the deeper layers, before the nutrients are entirely consumed (Lévy et al., 2001). Much less is known about the role of fine scales on phytoplankton diversity. However, the effect of fine-scale oceanic features on phytoplankton diversity has been predominantly studied using numerical simulations (Clayton et al., 2013; Barton et al., 2014; Lévy et al., 2015; Soccodato et al., 2016), while in situ sampling remains challenging because of the ephemeral nature of the dynamical structures. That is why the few targeted in situ sampling experiments have been principally performed in coastal upwelling regions (Ribalet et al., 2010) and in boundary currents (Clayton et al., 2014, 2017), which generate persistent fronts. These frontal structures associated with intense horizontal transport and vertical velocities (Allen and Smeed, 1996; Rudnick, 1996), can create physical barriers in the surface ocean, separating water masses and phytoplankton (Bower and Lozier, 1994). Furthermore, since the work of Yoder et al. (1994) these regions are well-known to be sites of high biological productivity.

More recently the numerical simulations of Barton et al. (2010) have shown that the fine scales may also be hotspots of phytoplankton diversity. In addition, exploiting remote sensing observations, d'Ovidio et al. (2010) have suggested that fronts play a key role in the generation of fluid dynamical niches, by segregating water patches with specific physical and chemical characteristics for time scales long enough for the emergence of a local phytoplanktonic community. It has then been argued that these features, which eventually mix together, can precondition biodiversity hotspots (De Monte et al., 2013; Soccodato et al., 2016). This scenario of a higher phytoplanktonic diversity driven by finescale fronts has recently received some empirical evidence by the comparison of molecular data of diatom diversity with satellite-derived front detections (Busseni et al., 2020).

However, the large majority of these works have focused on the extreme situations occurring in boundary currents, whose intense fronts and dramatic contrasts in water properties are not representative of the global ocean. On the contrary, vast oceanic

regions are dominated by weak fronts, which are continuously created, moved, and dissipated, and separate water masses with similar properties. Whether fine-scale fronts maintain their driving role on phytoplankton diversity also in these weaker regions remains therefore an open question which has been largely neglected, due to the difficulty of performing in situ experiments over these short lived features. In this study, we attempt to address this issue in a case study, focusing on a moderately energetic front, and to relate its role on the distribution of phytoplankton groups. Compared to more energetic regions, the effect of fine-scale on biogeochemical processes is non predictable here. This leads to the following question: is the physical forcing induced by horizontal and vertical motions sufficiently strong to segregate phytoplankton groups effectively in this moderately energetic region?

New sampling strategies were required to track the fine-scale ephemeral structures. Using remote sensing and numerical simulations to define the sampling strategy, field studies such as LatMix (Shcherbina et al., 2015), AlborEx (Pascual et al., 2017) and LATEX (Petrenko et al., 2017) have demonstrated that individual fine-scale features can be targeted experimentally. While these past campaigns focused mainly on physical processes, recent progress in biogeochemical sensors now makes the study of physical–biological coupling easier, including in moderately-energetic and oligotrophic areas. During the BLUE-FIN-13 cruise, Mena et al. (2016) studied the picophytoplankton distribution in a haline front formation area in the Balearic Sea. More recently, the OSCAHR cruise in the Ligurian Sea was specially designed to combine high-resolution measurements of both physical and biological variables (Doglioli, 2015). Exploiting the large data set collected during this cruise, Marrec et al. (2018) and Rousselet et al. (2019) have shown the influence of physical dynamics in controlling the spatial distribution of phytoplankton through an eddy structure, highlighting the close relationship between fine scale dynamics and the distribution of phytoplankton. In the southwest Pacific ocean, high-resolution biological sampling during OUTPACE (Moutin and Bonnet, 2015) and TONGA (Guieu and Bonnet, 2019) has shown the influence of fronts in controlling the spatial distribution of bacteria and phytoplankton (Rousselet et al., 2019; Benavides et al., 2021). Recently, the methodological developments in underway nitrogen ($N_2$) fixation measurements allowed to capture a rapid shift of the diazotrophic community in the North Atlantic ocean (Tang et al., 2020).

Following a satellite-based adaptive and Lagrangian sampling strategy, a high-resolution coupled physical–biological sampling was performed during the PROTEVSMED-SWOT cruise in the southwestern Mediterranean Sea, south of the Balearic Islands (Dumas, 2018; Garreau et al., 2020). This cruise was operated in the context of the preparation of the new satellite mission Surface Water and Ocean Topography (SWOT; https://swot.jpl.nasa.gov/,https://swot.cnes.fr, last access: July 16, 2021). This mission is dedicated to provide ocean topography and surface current at an unprecedented resolution, in particular during its fast sampling phase when SWOT will sample some key areas called "cross-overs" twice per day, offering a suitable opportunity to study the physical–biological fine-scale coupling (Morrow et al., 2019; d'Ovidio et al., 2019). The PROTEVSMED-SWOT cruise took place under one of these future cross-overs in the Mediterranean Sea. The large data set obtained during the cruise has been combined with another data set collected during the simultaneous and coordinated Spanish PRE-SWOT cruise (Barceló-Llull et al., 2018).

In this paper, we first describe the hydrodynamics of the area of study, with a focus on the vertical velocities estimated through the omega equation, and we identify the water masses residing in the region during the cruise. Then, the distribution of various groups of phytoplankton, and of fluorescent dissolved organic matter (FDOM), is presented in relation to the fine-scale dynamics. An advanced statistical analysis is performed to highlight the physical–biological coupling objectively.

## 2    Materials and methods

### 2.1    Satellite-based adaptive sampling strategy

PROTEVSMED-SWOT took place onboard the RV *Beautemps-Beaupré* between 30 April and 18 May 2018. During the cruise, an adaptive Lagrangian sampling strategy was possible thanks to SPASSO (Software Package for an Adaptive Satellite-based Sampling for Oceanographic cruises ; https://spasso.mio.osupytheas.fr, last access: July 16, 2021). Various satellite datasets were used during PROTEVSMED-SWOT. Sea surface temperature (SST, levels 3 and 4, 1 km resolution, not shown in this study) and chlorophyll $a$ concentrations ([Chla], level 3, 1 km resolution) were provided by CMEMS (Copernicus Marine Environment Monitoring Service, https://marine.copernicus.eu, last access: July 16, 2021). Moreover ocean color composite maps were provided by CLS with the support from CNES. They are constructed with a simple weighted average over the five previous days of the data issued from the Suomi/NPP/VIIRS sensor. The altimetry-derived geostrophic velocities from AVISO (Archiving, Validation and Interpretation of Satellite Oceanographic) database were exploited to extract near real-time daily maps. These were also used to derive the finite size Lyapunov exponents (FSLEs). The FSLE analysis permits to identify biogeochemical regions of potential interest. Indeed, FSLEs values often form continuous lines, or ridges, which are used to identify regions of enhanced strain that are to be expected near frontal zones. The first study that showed the interest of using FSLE-derived fronts for biogeochemical studies was probably Lehahn et al. (2007). Before, Abraham and Bowen (2002) have been the first to apply the Lyapunov exponent technique (although finite-time, not finite-size) to the ocean, in turn borrowing some ideas from dynamical system theory (see in particular Boffetta et al., 2001). A review on the FSLE and other satellite-based Lagrangian techniques can be also found in Lehahn et al. (2018); Hu and Zhou (2019). Since several years, this strategy has already been tested, either in post-cruise or real-time analysis, during many campaigns (e.g., Smetacek et al., 2012; d'Ovidio et al., 2015; Rousselet et al., 2018; De Verneil et al., 2019; Benavides et al., 2021). In the present work, FSLE have been obtained by time-integrating trajectories following the algorithm of d'Ovidio et al. (2004), after a 30 day backward integration.

SPASSO was used to follow both the temporal and spatial variability of the horizontal fine-scale features of interest. SPASSO combines satellite-derived currents, SST and [Chla], to provide maps of dynamical and biogeochemical structures in both Near Real Time (NRT) and Delayed Time (DT). During the cruise the analysis of these maps suggested the presence of two different regions, characterized by their different surface [Chla] (Fig. 1). Consequently, these two regions have been sampled along a dedicated route of the ship, represented in purple and in orange on Fig. 1. A special attention was paid to adapt the temporal sampling in these different water masses to the biological time scales, i.e. trying to catch the diurnal cycle. Because of the shape of the ship track, in the following we refer to these areas as "West-East (WE) hippodrome" (in orange on Fig. 1) performed

from 8 May 15:30 to 10 May 17:30 UTC, and "North-South (NS) hippodrome" (in purple on Fig. 1) performed between 11 May 02:00 and 13 May 08:30 UTC.

## 2.2 In situ measurements

Physical and biological variables (horizontal velocities, temperature, salinity, and abundances of the different phytoplankton functional groups) were measured at high frequency all along the route. In situ systems include a Seasoar deployed at sea, a
vessel-mounted Acoustic Doppler Current Profiler (VMADCP), a thermosalinograph (TSG) and a flow cytometer installed on board. The Seasoar is a towed undulating vehicle capable of achieving undulations from surface down to 400 m. Two Sea-Bird SBE-9 (with SBE-3 temperature and SBE-4 conductivity sensors) instruments mounted on either side of the Seasoar enables the simultaneous measurements of temperature, salinity (from conductivity) and pressure. The conductivity and temperature data were lag-corrected to reduce salinity spiking with the methodology developed by Lueck and Picklo (1990), Morison et al.
(1994) and Mensah et al. (2009), before conversion to absolute salinity $S_A$ and conservative temperature $\Theta$, in accordance with TEOS-10 standards (McDougall et al., 2012). In the following, temperature and salinity refer to absolute salinity and conservative temperature. Horizontal velocities between 19 m and 253 m have been measured with a VMADCP operating at 150 kHz. VMADCP data treatment was performed with the Matlab software Cascade V.7 (https://www.umr-lops.fr/en/Technology/Software/CASCADE-7.2, last access: July 16, 2021). The sea surface temperature and salinity have been measured
continuously along the ship route by the underway TSG. The TSG was equipped with two sensors: i) a CTD sensor SeaBird Electronics SBE 45, installed in the wet lab, connected to the surface water, and pumping continuously seawater at 3 m depth. ii) a temperature sensor SBE 38 installed at the entry of the water intake.

An automated CytoSense flow cytometer (CytoBuoy b.v.) was installed on board and connected to the seawater circuit of the TSG, to perform scheduled automated sampling and analysis of phytoplankton (Thyssen et al., 2009, 2015). The instrument
contains a sheath fluid made of 0.1 μm filtered seawater which stretches the sample in order to separate, align and drive the individual particles (i.e. cells) through a light source. This light source is made of a 488 nm laser beam. When the particles cross the laser beam, they interact with the photons. Several optical signals are recorded for each single particle: the forward angle light scatter (FWS) and 90° side-ward angle scatter (SWS), related to the size and the structure (granularity) of the particles. Two signals of fluorescence induced by the light excitation were also recorded, a red fluorescence (FLR) induced
by chlorophyll $a$ and an orange fluorescence (FLO) induced by the phycoerythrin pigment. Two distinct protocols have been run sequentially every 30 min, in order to process the 1164 samples. The first protocol (FLR6) has a FLR trigger threshold fixed at 6 mV and is able to analyze a volume of 1.5 cm$^3$. It was dedicated to the analysis of the smaller phytoplankton. For instance, *Synechococcus* were optimally resolved and counted with this protocol. The second protocol (FLR25), targeted nanophytoplankton and microphytoplankton with FLR trigger level fixed at 25 mV and an analyzed volume of 4 cm$^3$. The
data were acquired thanks to the USB software (Cytobuoy b.v.) but analyzed with the CytoClus software (Cytobuoy b.v.). The combination of the various variables recorded by the flow cytometer exhibits various clusters of particles (cells), which abundance (cells per cubic centimetre) and average variable intensities are provided by the CytoCLUS software. The latter

generates several two-dimensional cytograms (e.g. Fig. 6, see section 3.3 for explanation of the group identification) of retrieved information from the 4 pulse shapes curves (FWS, SWS, FLO, FLR) obtained for every single cell.

On 5 May 2018, a SeaExplorer glider, manufactured by Alseamar (codename: SEA003), was deployed at sea. After a short transit, it performed a route approximately parallel to the NS hippodrome (red track on Fig. 1). This glider, set to dive down to 650 m depth, was equipped with a pumped conductivity-temperature-depth sensor (Seabird's GPCTD) from which the conservative temperature ($\Theta$), the absolute salinity ($S_A$) and the density anomaly referenced to the surface ($\sigma_0$) were derived using TEOS-10 toolbox (McDougall and Barker, 2011). This GPCTD was also equipped with a dissolved oxygen ($O_2$) sensor

(Seabird's SBE-43F) to measure oxygen concentrations. The glider also embarked a WET Labs ECO Puck FLBBCD for measurements of i) [Chla] fluorescence (targeting excitation and emission wavelengths at $\lambda$Ex/$\lambda$Em: 470/695 nm), converted into Chla concentrations (in microgram per litre), ii) backscattering at 700 nm (BB700), and iii) a FDOM fluorophore, namely the humic-like fluorophore or peak C in the Coble (1996)'s classification ($\lambda$Ex/$\lambda$Em: 370/460 nm), expressed in microgram per litre equivalent quinine sulfate units (microgram per litre QSU). Finally, the SeaExplorer was also equiped with two

MiniFluo-UV fluorescence sensors (hereafter called MiniFluo) for the detection of various FDOM fluorophores (Cyr et al., 2017, 2019). In this study, the MiniFluo-1 was used for the detection of tryptophan-like fluorophore ($\lambda$Ex/$\lambda$Em: 275/340 nm), while the MiniFluo-2 was used for the detection of tyrosine-like fluorophore ($\lambda$Ex/$\lambda$Em: 260/315 nm). Tryptophan- and tyrosine-like fluorophores, referred to as peak T and peak B, respectively, in the Coble (1996)'s classification, are amino acid-like components commonly found in the marine environment, and generally associated with autochthonous biological processes

(see review by Coble et al. (2014)). Here, fluorescence intensities of tryptophan- and tyrosine-like fluorophores are provided in relative unit (RU) and are not converted into mass concentration (microgram per litre) (Cyr et al., 2017, 2019). Glider observations were processed with the Socib glider toolbox (Troupin et al., 2015) for cast identification and geo-referencing.

## 2.3    Vertical velocity estimation

The vertical velocity has been diagnosed solving the so-called quasi-geostrophic (QG) omega equation (Hoskins et al., 1978;
Tintoré et al., 1991; Allen and Smeed, 1996, Eq. (1)):

$$\nabla^2(N^2 w) + f^2 \frac{\partial^2 w}{\partial z^2} = 2\nabla \cdot \boldsymbol{Q} \tag{1}$$

where $w$ is the vertical component of the velocity field and $\boldsymbol{Q}$ is the vector determined by horizontal derivatives of water density and horizontal velocity (Hoskins et al., 1978; Giordani et al., 2006, Eq. (2)):

$$\boldsymbol{Q} = \left( \frac{g}{\rho_0} \frac{\partial \boldsymbol{V_g}}{\partial x} \nabla \rho, \frac{g}{\rho_0} \frac{\partial \boldsymbol{V_g}}{\partial y} \nabla \rho \right) \tag{2}$$

with $\boldsymbol{V_g}$ the geostrophic horizontal velocity vector, $\rho$ the density, $\rho_0$ a reference density equal to $1025 \, \mathrm{kg\,m^{-3}}$, $g$ the gravitational acceleration, $f$ the Coriolis parameter (considered constant and computed at the mean latitude of the area), and $N^2$ the Brunt–Väisälä frequency. The QG theory is valid for low Rossby numbers, a condition that is satisfied in this study.

High-resolution in situ data are necessary to solve Eq. (1). In this work, $\sigma$ is obtained from Seasoar CTD measurements (Fig. A1) and the geostrophic component of the horizontal velocity has been estimated from the measurements performed with the VMADCP as in the work of Barceló-Llull et al. (2017). Following Allen et al. (2001)'s suggestions to preserve as much as possible synopticity, four transects of the NS hippodrome (see Fig. 1) have been selected between 11 May and 12 May 2018, to obtain a "butterfly" design as in Cotroneo et al. (2016) and Rousselet et al. (2019). The $\sigma$, u and v fields have been interpolated onto a 3D grid, using objective analysis (Le Traon, 1990; Rudnick, 1996). The horizontal grid resolution is $0.9 \, \text{km} \times 0.9 \, \text{km}$ and the vertical resolution is $6 \, \text{m}$ (from $19 \, \text{m}$ to $253 \, \text{m}$ depth). We have followed the method described by Rudnick (1996), using the scripts freely downloadable from his web page at the address http://chowder.ucsd.edu/Rudnick/SIO_221B.html, last access: 10 February 2021. The in situ data are considered to be composed of a mean value, a fluctuation and some noise including both smaller scale variability and instrumental error. The fluctuation part of the field's statistics is assumed to have a decorrelation length scale of $20 \, \text{km}$ in both the x and y directions, with a structure orientation of $18.4°$ from north. The correlation length scale has been chosen by analyzing the auto-covariance matrix of the $\sigma$ field and performing several sensitivity tests. The noise-to-signal ratio is assumed to be 0.05 as in Rudnick (1996). The interpolated fields are shown superposed to the in situ measurements (see Fig. A1 in Appendices). The ageostrophic component of the velocity measured by the VMADCP is then removed and Eq. (1) is solved with an iterative relaxation method and constrained by Dirichlet boundary conditions ($w = 0$) as in the case of the front studied by Rudnick (1996). To minimize the effect of the imposed boundary conditions, only data with an error on the objective mapping of $\sigma \leq 0.0025$ have been then considered.

## 3  Results

### 3.1  Hydrodynamics

At first, we describe the hydrodynamic conditions encountered during PROTEVSMED-SWOT in order to characterize the area. For simplicity, two transects only, representative of each of the two hippodromes, are described: a first transect, on the WE hippodrome, performed from 9 May 16:50 to 23:45 UTC is referred to as the WE transect (Fig. 2a). The second one performed on the NS hippodrome, from 11 May 02:00 to 08:40 UTC, is referred to as the NS transect (Fig. 2b). Note that we obtained similar results for the other transects (see Fig. A2 in Appendices). The horizontal velocities have been measured by VMADCP at $25 \, \text{m}$ along both WE and NS transects, and superimposed to the FSLE field for the corresponding date. The intensity and the direction of the current vary along the transects. Around 38° N 20', a zonal fine-scale feature, slowly evolving, is present in the altimetry-derived FSLE field and confirmed by the VMADCP data. Indeed, two FSLE features cut the transect at this latitude, exactly where the horizontal current directions change drastically. The WE transect shows a larger current variability than the NS transect, due to its alignment with the fine-scale structure. However, an FSLE feature cuts this transect just above 38° N 20' and, at this point, the current begins to change and turns to the North-East. The FSLE features and the variation of current direction are likely induced by the presence of a fine-scale structure, in this case a front.

Figures 3a and 3b show the vertical velocities estimated at $25 \, \text{m}$ and $85 \, \text{m}$ depths for the NS hippodrome. The area is characterized by three main features: two upwelling cells (positive values) separated by a downwelling cell (negative values)

located between 38° N 30' and 38° N 36'. Another smaller downwelling patch is present in the south-east of the sampling area. The intensities of these vertical motions, ranging from 2 to 8 $10^{-5}$ m s$^{-1}$ (corresponding to 1.7 to 6.9 meters per day), are stronger in the intermediate layer.

## 3.2 Hydrology

The typical southwestern Mediterranean water masses are observed in the $\Theta$–$S_A$ diagrams of the Seasoar CTD data of the WE and NS transects (Fig. 4a and Fig. 4c, respectively). A clear separation in two different water masses appears at the surface. In order to identify where these two water masses separate objectively, we first distinguish surface from intermediate waters by the isopycnal at 28.45 kg m$^{-3}$ on the $\Theta$–$S_A$ diagrams. Then, an iterative method with a granularity of 0.05° in longitude(latitude) along the WE(NS) transect has been used to calculate the means of the two surface water masses in terms

of salinity and temperature, $(\overline{S_{A1}}, \overline{\Theta_1})$ and $(\overline{S_{A2}}, \overline{\Theta_2})$, and the difference in terms of mean salinity, $|\overline{S_{A1}} - \overline{S_{A2}}|$. Then, the best separation between the two surface water masses corresponds to the longitude(latitude) along the WE(NS) transect where the maximal difference $|\overline{S_{A1}} - \overline{S_{A2}}|$ is found. Table 1 summarizes the maximal differences calculated for the transects of the WE and NS hippodromes performed with the Seasoar and the associated localisations of the best separation. Only one transect of the WE hippodrome is shown because we deplore a lack of data in the Seasoar measurement during the other transects of this

hippodrome. In Table 1, along the WE transect the surface water masses separate at longitude 4° E 06', while along the NS transect separate at the latitude 38° N 32', as also indicated by the triangles on the corresponding figures (Fig. 4b and Fig. 4d). Note that for the other meridional transects the estimated separation varies of few minutes only (Table 1). $\Theta$–$S_A$ diagrams (Fig. 4a and Fig. 4c) and their associated maps (Fig. 4b and Fig. 4d), correspond to the WE and the NS transects described in the previous section 3.1. The maps shown in Fig. 4b and Fig. 4d indicate the geographical positions of the two types of AW,

using the same color code as in the $\Theta$–$S_A$ diagrams. Interpretation of these data is the following. The surface layer is occupied by Atlantic Waters (AW) with different residence times in the Mediterranean Sea. We refer to them as "younger AW" (in light blue) and "older AW" (in dark blue). The "younger AW" corresponds to AW entered more recently in the Mediterranean basin and characterized by a salinity between 37 g kg$^{-1}$ and 38 g kg$^{-1}$, while the "older AW" is characterized by a higher salinity. Other authors refer to this water as "local AW" (Barceló-Llull et al., 2019) or "resident AW" (Balbín et al., 2012). The "younger

AW" is located at the west and the south of the WE and NS transects, respectively. Moreover, the separation between the two types of AW is in agreement with the localisation of the front identified by the FSLE and the change in the current direction (Fig. 2). Below the surface layer, the intermediate water is more homogeneous and is characterized by a temperature of 13–14 °C and a salinity of 38–38.5 g kg$^{-1}$ (in pink on Fig. 4a and Fig. 4c).

The SeaExplorer glider has also performed temperature and salinity measurements along a transect parallel and slightly

west of the NS hippodrome (see Fig. 1). The $\Theta$–$S_A$ diagrams of the glider data (Fig. A3a and Fig. A3b) confirm the presence of the two surface water masses mentioned above, in particular during the outward route (Fig. A3a). During the glider return route (Fig. A3b) the surface water masses begin to be more homogeneous than during the outward route and the NS Seasoar transect (Fig. 4c). This fact can be explained because the transects completed with the Seasoar were realized within a few hours, while the glider transects lasted several days. Moreover, the deeper glider sampling has allowed to detect another thermohaline

signature, with temperature values about 13 °C and salinity about 38.5 $\mathrm{g\,kg^{-1}}$, corresponding to the Western Mediterranean Deep Water (WMDW) as found by Balbín et al. (2012) and Barceló-Llull and Pascual, in revision for Front. Mar. Sci.

The vertical extension of the surface and intermediate water masses can be observed plotting the Seasoar data (conservative temperature, absolute salinity and density) as vertical sections along the WE and NS transects (Fig. 5). The warm surface layer with temperature greater than 15 °C extends until about 100 m, on Fig. 5a and Fig. 5d. This layer is also characterized by a salinity between 37.5 $\mathrm{g\,kg^{-1}}$ and 38 $\mathrm{g\,kg^{-1}}$ (Fig. 5b and Fig. 5e) and, as a consequence, by the lowest density (Fig. 5c and Fig. 5f). Note that this surface layer is more apparent along the transects of the WE (Fig. 5a, b and c), than along the NS transect. Below 100 m depth there is the intermediate water layer. Looking at gradients along the temperature and salinity transects, we can clearly identify the separation between the two types of AW at longitude 4° E for the WE transect (Fig. 5a and 5b) and latitude 38° N 30' for the NW transect (Fig. 5d and Fig. 5e), as obtained with the iterative method. However, this separation is less apparent along the two density transects (Fig. 5c and Fig. 5f).

### 3.3 Characterization and distribution of phytoplankton by flow cytometry

Up to nine phytoplankton groups were optically resolved by flow cytometry (Fig. 6), thanks to their light scatter (forward scatter, FWS, and sideward scatter, SWS) and their red (FLR) and orange (FLO) fluorescence intensities, as in the work of Thyssen et al. (2015) and Marrec et al. (2018). We have called these groups with the conventional names used by flow cytometrists, i.e., some groups related to the taxonomy (*Synechococcus*, Cryptophytes) while the others related to a range of size (picoeukaryotes, nanoeukaryotes) according to Sieburth et al. (1978). Indeed, the first group corresponds to *Synechococcus* (Syn on Fig. 6c), which is a prokaryotic picophytoplankton. We distinguish it from the other picophytoplanktonic groups, because it was unambiguously put in evidence by flow cytometry thanks to its higher FLO intensity compared to the FLR intensity (Fig. 6c), induced by the presence of phycoerythrin pigments. A first eukaryotic picophytoplankton group (Pico1) shows lower FLR and FLO intensities than *Synechococcus* (Fig. 6c). Two other groups of picophytoplankton (Pico2 and Pico3) exhibit higher FWS, SWS and FLR intensities than Pico1 (Fig. 6d). The last group of picophytoplankton (PicoHFLR) is characterized by a high FLR signal induced by Chla. Two distinct nanophytoplankton groups (SNano) and (RNano) were defined according to their high FLR and FLO intensities (Fig. 6a and Fig. 6c). SNano have a high SWS/FWS ratio and higher SWS intensities than RNano (Fig. 6b) and Fig. 6a). Finally, microphytoplankton (Micro) and Cryptophytes (Crypto) exhibit high FLR and FLO intensities (Fig. 6c). Cryptophytes can belong to the pico- or nanoeukaryotes but have also been discriminated from the red-only fluorescing pico- or nanoeukaryotes based on their orange fluorescence induced by phycoerythrin, like *Synechococcus*. The size and the abundances of these nine groups have been summarized in Table 2.

Figures 7 and 8 show the surface abundances of the various phytoplankton groups, along the WE and the NS transects. *Synechococcus* (Fig. 7a and Fig. 8a), Pico1 (Fig. 7b and Fig. 8b) and Pico2 (Fig. 7c and Fig. 8c) present a similar distribution pattern. High abundances around 2.2–3.0 $10^4\ \mathrm{cells\,cm^{-3}}$, $\geq 1.1\ 10^3\ \mathrm{cells\,cm^{-3}}$, and $\geq 1.7\ 10^3\ \mathrm{cells\,cm^{-3}}$, recorded respectively for *Synechococcus*, Pico1, and Pico2 are located at the western and southern parts of the front, along with the WE and the NS transects. On the other side of the front, their abundances are lower, ($\leq 1\ 10^4\ \mathrm{cells\,cm^{-3}}$ for *Synechococcus* and $\sim 500$ and $\sim 900\ \mathrm{cells\,cm^{-3}}$ for Pico1 and Pico2, respectively). RNano abundances (Fig. 7g and Fig. 8g) present a similar distribu-

tion with these groups along the NS transect, with high abundances (450–500 $\text{cells cm}^{-3}$) located at the southern part of the

front. However, the distribution of this latter is less clear along the WE transect. Pico3 (Fig. 7d and Fig. 8d), Microphytoplankton (Fig. 7f and Fig. 8f), and SNano (Fig. 7h and Fig. 8h) abundances vary between 100–500 $\text{cells cm}^{-3}$, 8–18 $\text{cells cm}^{-3}$, and 300–500 $\text{cells cm}^{-3}$, respectively, and present an opposite distribution compared to the other previous groups. Indeed, higher abundances are found in the eastern and northern parts of the front. PicoHFLR (Fig. 7e and Fig. 8e) and Cryptophytes abundances (Fig. 7i and Fig. 8i) ranged from 160–280 $\text{cells cm}^{-3}$ and 10–30 $\text{cells cm}^{-3}$ respectively. However, these latter

exhibited a less obvious pattern between the two sides of the front. Overall, the distribution of the abundances of the various phytoplankton groups evidenced by flow cytometry on either side of the front (except for these two last groups) fits well with the hydrodynamic and hydrological observations.

### 3.4 Statistical analysis

Surface temperature and salinity data measured with the TSG were merged with abundance data of the 9 phytoplankton groups

at each cytometry sampling point along the WE and NS hippodromes. Thus, the final data set consists of 11 variables, each variable containing 215 observations. To deal with this large multivariate data set a principal component analysis (PCA) was applied.

The PCA consists of summarizing the information contained in the data set by replacing the initial variables with new synthetic variables (called principal components), linear combinations of the initial variables and uncorrelated two by two.

Applied to our data set, the PCA points out that the first three components explain 36.7 %, 18.1 % and 13 % of the total variance of the data, respectively. The following statistical analysis focuses on these three components, representing 67.8 % of the total variance of the data. Figure 9a shows positively correlated variables grouped together in the first factorial plane: salinity ($S_A$) with microphytoplankton (Micro); *Synechococcus* (Syn) with picophytoplankton Pico1 group; picophytoplankton Pico2 group and nanophytoplankton RNano group; temperature ($\Theta$) with picophytoplankton Pico3 group and nanophytoplankton

SNano group. However, Cryptophytes (Crypto) group and picophytoplankton PicoHFLR group are not as well correlated with the other variables compared to the other groups of phytoplankton.

The K-medoid algorithm, described by Hartigan and Wong (1979) and Kaufman and Rousseeuw (1987), is an-other method to represent the various aspects of the data structure. This algorithm divides M points in N dimensions into K groups (or clusters). A cluster is an object for which the average dissimilarity to all the data is minimal. In our case of study, the K-medoid

algorithm splits the 215 points (i.e., observations) into three clusters (Fig. 9b). Each point of a cluster shows a high degree of similarity with the others points of the same cluster. The three clusters are well separated from each other. Only a few points of the black and red clusters are difficult to disentangle. Finally, the average of each variable called local average and the global average have been calculated for each cluster to show the contribution of each variable to a cluster. The most discriminating variables for each cluster (Table 3) have also been determined with the standard deviation.

Figure 9c is a spatio-temporal representation of the three main clusters obtained by the K-medoid algorithm. The WE hippodrome from 8 to 10 May is characterized by the presence of the black cluster in the east and the red cluster in the west. The NS hippodrome starts the 11 May with the red cluster present in the south and the black cluster in the north. The latter

remains dominant in the north for the remaining of the sampling. In the south, the red cluster is gradually replaced by the blue cluster, except for a few points on the 13 May.

Figure 9d displays the geographical distribution of the three clusters superposed on the FSLE field. It evidences a general good agreement between the shifts of the clusters and the FSLE maxima. In particular, the separation between the black and red clusters at around 4° E on the WE hippodrome corresponds to the two FSLE maxima crossing this longitude at 38° N 40' and 38° N 20'. On the NS hippodrome, the black cluster is separated from the two others at about 38° N 20' where zonally-disposed FSLE maxima cross the vessel route. To summarize, the black cluster dominates in the north of the sampled area,

while in the south the red cluster is dominant at the beginning but then is replaced by the blue cluster. The separation between the clusters matches the distribution of the maxima of the FSLE field, and can be explained by the fine-scale dynamics. Given their ephemeral nature the fine-scale structures, the temporal evolution of the distribution of the red and blue cluster is probably due to the displacement over time of the frontal structure during the cruise.

## 4 Discussion

### 4.1 Physical properties of the front

During the PROTEVSMED-SWOT cruise, the satellite-derived surface [Chla] showed contrasted values between the south-west and the north-east of the studied area (Fig. 1). The analysis of the ocean color images combined with the altimetry-derived FSLE field, have been associated with the strong variation of current direction observed in the horizontal velocities measured by VMADCP data, allowing the identification of a frontal area located at the latitude of about 38° N 30' and between 3° E and

4° E of longitude (Fig. 2).

Our estimation of vertical velocities with the omega equation method allowed to investigate the vertical dynamics in the frontal area (Fig. 3). Several previous studies have shown the importance of fine-scale dynamics in generating vertical velocities (e.g., Rudnick, 1996; Lapeyre and Klein, 2006; Mahadevan and Tandon, 2006; Balbín et al., 2012). Although the ship route was designed mainly for the cytometry sampling across the front, we obtained a spatial grid sufficiently regular to apply

the omega equation method, as recommended by Allen et al. (2001). In the area where our sampling with the Seasoar overlaps with the CTD casts performed on a 10 km spaced regular grid by Barceló-Llull et al. (2018), the respective estimations of the vertical velocity field are in good agreement (Barceló-Llull and Pascual, in revision for Front. Mar. Sci.). As our sampling extends further south with respect to the one of Barceló-Llull et al. (2018), we were also able to observe the shift in the current direction, from eastward to westward, and the associated vertical recirculation pattern. Vertical velocities associated with the

front are in the order of few meter per day. These values are of the same order of magnitude as the velocities associated with a frontal structure reported by Balbín et al. (2012) in the north-west of the Balearic islands, and by Ruiz et al. (2019) in the Alboran Sea.

The data from the CTD sensors mounted on the Seasoar towed fish and on the SeaExplorer ocean glider identify a rapid shift

between two different types of surface water masses on the $\Theta$–$S_A$ diagrams (Fig. 4 and Fig. A3). These water masses are AW

at different stages of modification. Indeed, the AW enters into the Mediterranean Sea through the Strait of Gibraltar and then forms an anticlockwise circulation along the continental slope of the western Mediterranean basin caused by the combination of the Coriolis effect and the topographical forcing (Millot, 1999; Millot and Taupier-Letage, 2005; Millot et al., 2006). In the south-west of the basin, this circulation is dominated by the Algerian Current (AC), which can form meanders and mesoscale eddies due to baroclinic and barotropic instabilities (e.g., Millot, 1999). These eddies spread over the basin and join the region south of Balearic Islands, carrying with them the AW recently entered in the Mediterranean that we define "younger AW". In this region, the "younger AW" encounters the "older AW", i.e. the AW modified by cooling and evaporation while progressing along the northern part of the western Mediterranean basin (Millot, 1999; Millot and Taupier-Letage, 2005). The presence of this water has already observed by Balbín et al. (2012) and Barceló-Llull et al. (2019) which refers to it as "resident" or "local" AW, respectively, to indicate the colder and saltier AW of the Balearic Sea.

During our cruise, the "older and younger AW" are separated by a frontal area (Fig. 4). This separation between these two AWs is also clearly visible in the first 100 m of the water column, on the depth sections of temperature and salinity (Fig. 5). López-Jurado et al. (2008) and Balbín et al. (2012, 2014), have observed similar results and have developed some hypotheses explaining the seasonal and inter-annual variability of the location and intensity of the front between these two AWs. According to Balbín et al. (2014), the presence or absence of Intermediate Water in the Mallorca and Ibiza channels at the beginning of the summer, determines the meridional position of the front.

## 4.2 Biogeochemistry

Vertical sections of [Chla] and oxygen [$O_2$], issued from the SeaExplorer glider have been measured along the outward route (Fig. A4a and Fig. A4b) and the return route (Fig. A4c and Fig. A4d) parallel to the NS hippodrome, in the first 250 m of the water column. These glider vertical sections show patterns that can be explained by the vertical movement of the water masses. A deepening of the maximum [Chla] is present in the north of 38° N 30' along the outward route (Fig. A4a) and the return route (Fig. A4c) of the SeaExplorer glider, corresponding to the "older AW" identified on Fig. 4d. Similar patterns also observed in the [$O_2$] data confirm a possible downwelling of the surface waters at the front location (see Fig. 3), but also indicate a deeper oxygenation of the "older AW".

The vertical sections of tyrosine- and tryptophan-like fluorophores (respectively peak B and peak T) fluorescence intensities revealed distribution patterns very close to that of [Chla] for tyrosine-, and very close to that of [$O_2$] for tryptophan-like fluorophore (data not shown). These results highlight the strong coupling between hydrology, phytoplankton activity and DOM concentration in this area. Indeed, tryptophan- and tyrosine-like fluorophores are recognized to have an autochthonous origin (Coble et al., 2014), being produced through the activity of autotrophic and heterotrophic plankton organisms, in particular phytoplankton and heterotrophic bacteria (Stedmon and Cory, 2014), and are known to be indicators of bioavailable/labile DOM (C and N) (Hudson et al., 2008; Fellman et al., 2009). Even though phytoplankton activity is considered a source of tryptophan- and tyrosine-like fluorophores (Determann et al., 1998; Stedmon and Markager, 2005; Romera-Castillo et al., 2010), bacterial degradation appears to be a source, but also a sink for these fluorophores, depending on the availability in nutrients (Cammack et al., 2004; Nieto-Cid et al., 2006; Biers et al., 2007). These similarities in the profiles of [Chla] and

[$O_2$] were confirmed by correlation analyses (Fig. A5), which pointed out a very highly significant linear positive correlation between [Chla] and tyrosine-, and between [$O_2$] and tryptophan-like fluorophores when considering all glider data for the two transects from 5 to 200 m depth (r = 0.88 and 0.84, n ≈ 32595, p < 0.0001). The fact that tyrosine-like fluorophore was rather associated with [Chla] and tryptophan-like with [$O_2$] reveal that these two fluorophores were probably not issued from the same phytoplankton groups. Moreover, it seems that tryptophan would be more susceptible to be released by heterotrophic bacteria

(in addition to be released by phytoplankton) than would be tyrosine-like material (Hudson et al., 2008; Tedetti et al., 2012; Stedmon and Cory, 2014). The high contents in tryptophan- and tyrosine-like fluorophores found in the northern part of the transect ("older" AW) must be correlated with the abundances of Microphytoplankton at this place. Indeed, it has been recently shown that various groups of microphytoplankton might produce tryptophan- and tyrosine-like fluorophores (Romera-Castillo et al., 2010; Fukuzaki et al., 2014; Retelletti Brogi et al., 2020).

**4.3   Physical-biological coupling in the frontal area**

  The distribution of phytoplankton groups showed contrasted abundances across the front. Figure 10 summarized our results, providing a view of the physical forcing occurring in the frontal area and its effect on the distribution of phytoplankton groups. The south-west of the front corresponding to the localisation of the "young AW" is characterized by high abundances of *Synechococcus*, Pico1, Pico2 and RNano groups, whereas the north-east of the front associated with the "old AW" is dominated

by the other phytoplankton groups, i.e. Pico3, Microphytoplankton and SNano (Fig. 7 and Fig. 8).

  Although some studies led in western boundary currents have shown that fronts are sites of elevated phytoplankton diversity (Barton et al., 2010; Clayton et al., 2014), it is still unclear how the physical processes impact the phytoplankton distribution. That is why three mechanisms, i.e., advection, mingling and fertilization may be advanced as possible explanation for the distribution of phytoplankton abundances, as observed during our cruise.

First, the dominance of certain groups of phytoplankton appears well related with the type of AW. The statistical analysis proved very useful in synthesizing the physical and biological information and in identifying the relationship between some groups of phytoplankton and the hydrological water conditions objectively. For instance, on the principal component analysis (PCA), *Synechoccocus*, salinity and temperature are well correlated (Fig. 9). This correlation is in accordance with the study of Mena et al. (2016), which have already found higher abundances of *Synechococcus* in the "new" AW less salty than the

"resident AW", and this is also in agreement with Marrec et al. (2018) who have shown a dominance of *Synechococcus* in warm boundary waters. Previous studies have shown that fine-scale can form ecological niches, by segregating water masses long enough and thus creating favourable conditions for some phytoplankton groups locally well adapted which will become dominant (d'Ovidio et al., 2010; Perruche et al., 2011). d'Ovidio et al. (2010) have also showed that these fluid ecological niches can be advected and then can transport over a distance far apart from their origin the phytoplankton cells, creating

complex community distributions by enhancing the mingling of populations of phytoplankton. This theory has been deepened by Clayton et al. (2013) by global numerical simulations. According to them, dominant phytoplankton very well "locally" adapted can emerge in hotspot regions, whereas other phytoplankton called "immigrants" are sustained by physical transport. The Mediterranean spans a much reduced latitudinal gradient. In our study, the intensity of currents allows a slower advection

than the one by western boundary currents, and our cytometry measurements do not show presence or absence of specific groups. Nevertheless, our results highlight that fine-scale dynamics modulate the relative abundance of phytoplankton groups. We can assume for instance that the "young" AW could have transported some phytoplankton cells from the Algerian basin to the Balearic Islands. However, the front observed during our cruise can not be defined as an "hotspot" of diversity as in the work of Clayton et al. (2014, 2017), where fine-scale features advect phytoplankton ecotypes that are absent in an oceanic region to an other. In future cruises, we plan to perform taxonomic analyses such as metagenomics, that will allow us to detail the structure of the communities further and tell more about competition and immigration processes.

Second, we can point out the role of vertical velocity (see Fig. 3). Clayton et al. (2014, 2017) have highlighted the role of fine-scale circulation in regulating phytoplankton diversity, in particular for opportunistic species (as diatoms) fertilized by an important nutrient enrichment induced by a very intense vertical transport in boundary currents and coastal upwelling regions. In our case, we found vertical velocities associated to the front less intense than in the work of Clayton et al. (2014, 2017). This vertical transport can advect a sufficient quantity of nutrients in the euphotic layer, creating favorable conditions for phytoplancton groups well adapted to the oligotrophic conditions of the Mediterranean Sea. Indeed, previous modelling study of Lévy et al. (2001), have shown the significant impact of the enhancement of nutrient on the production of phytoplankton in an oligotrophic regime. In addition to a role on nutrient distribution, the vertical velocity can also directly convey phytoplankton cells and, as a consequence, play a key role in their distributions (Marrec et al., 2018). In order to disentangle these two mechanisms, additional both high-resolution and, since the oligotrophic Mediterranean conditions, high-precision nutrient measurements need.

## 5   Conclusion and perspectives

In conclusion, our adaptive Lagrangian strategy and high-resolution coupled physical–biological sampling has allowed to detect a fine-scale frontal structure and has highlighted its structuring effect on the surface phytoplankton community, in accordance with previous modelling studies. The originality of our work resides in the fact that we are able to show that less energetic fronts than those found in western boundary currents have also an impact on the phytoplankton distribution. This suggests that the physico-chemical contrasts induced by the horizontal stirring processes in moderately energetic fronts are sufficiently strong to spatially reflect into phytoplanktonic communities with different structures. Furthermore, our work offers an interesting explication concerning the fact that despite the Mediterranean Sea is an oligotrophic regime, this region is also characterized by a high diversity of phytoplankton. Because fine-scale processes and oligotrophic regions are predominant in ocean, our results shed a new light on the functioning of the global ocean.

In the future, a better understanding of the biogeochemical processes generating this observed fine-scale physical–biological coupling is needed. In particular, we plan to estimate and compare the growth rates of the various phytoplankton groups in the different water masses thanks to the data collected in a Lagrangian manner and applying a method similar to the one by Marrec et al. (2018). Moreover, the role of nutrient supply (bottom-up) and of zooplankton grazing (top-down) are also key factors to be considered in explaining the differences in abundances of the different phytoplankton groups in the different

areas visited during the cruise. To address these latter points, future experiments will need high-resolution (and also high-precision, considering the oligotrophy of the Mediterranean Sea) nutrient measurements coupled with zooplankton sampling and dedicated experiment about its grazing on the different phytoplankton groups. Finally, this study also shows how the satellite information is extremely useful for the design of the cruise sampling strategy and, then, for the on-shore post-cruise interpretation of the data.

The new satellite SWOT (https://swot.jpl.nasa.gov/, last access: July 16, 2021; https://swot.cnes.fr, last access: July 16, 2021) will provide ocean topography and surface current with a resolution one order of magnitude higher than present altimeters (Morrow et al., 2019). During the few months after its launch, the period called "fast sampling phase", the satellite will be on a special orbit that will overfly a portion of the global ocean, called "cross-overs", where high spatial resolution will be associated with high temporal resolution (d'Ovidio et al., 2019). The large data set collected during PROTEVSMED-SWOT represents precious new information on the SWOT cross-over area located in the south of the Balearic Islands. The work presented here thus paves the way to future cruises to be planned in this area during the SWOT period, which will be a unique opportunity to study the physical–biological fine-scale coupling in more detail.

*Code availability.* The LATEXtools can be openly available: https://people.mio.osupytheas.fr/~doglioli/latextools.htm, last access: July 16, 2021 (Doglioli et al., 2013).

The script for objective mapping can be openly available on Rudnick's web page: http://chowder.ucsd.edu/Rudnick/SIO_221B.html, last access: July 16, 2021.

*Data availability.* The data can be openly available: https://www.seanoe.org/data/00512/62352/, last access: July 16, 2021 (Dumas et al., 2018).

*Author contributions.* RT post-processed the in situ observations, performed the analysis of the results and leaded the writing of the manuscript. AMD and GG designed the Lagrangian experiment and collected the in situ data together with FD and PG. AAP, SB and FdO provided on land support to the sampling strategy. LI and MTh carried out the analysis of flow cytometry data. AP and BBL contributed to the vertical velocity analysis. FC, NB and MTe conducted the glider deployment and the processing of its data. All the authors discussed the results and contributed to the writing of the manuscript.

*Competing interests.* The authors declare that they have no conflict of interest.

*Acknowledgements.* This work was supported by the CNES in the framework of the project BIOSWOT-AdAC (https://www.swot-adac.org/, last access: July 16, 2021) will coordinate several cruises, and by the MIO Axes Transverses program (AT-COUPLAGE). The chlorophyll *a* product is produced by CLS. The authors thank the SHOM and the crew of the RV *Beautemps-Beaupré* for shipboard operations, M. Torner, the SOCIB and the crew of the RV *García del Cid* for their assistance with the glider operations, J.-L. Fuda for his help in Seasoar data treatment, L. Rousselet for discussion about vertical velocity estimations, and M. Goutx for the early discussions about the glider FDOM data. The Cytobuoy® flow cytometer was funded by the CHROME project, Excellence Initiative of Aix-Marseille University – A*MIDEX, a French "Investissements d'Avenir" program. SPASSO is operated and developed with the support of the SIP (Service Informatique de Pythéas) and in particular C. Yohia, J. Lecubin. D. Zevaco and C. Blanpain (Institut Pythéas, Marseille, France). The project leading to this publication has received funding from European FEDER Fund under project 1166-39417. This work was also supported by the French National program LEFE (Les Enveloppes Fluides et l'Environnement) (FUMSECK-vv project, PI S. Barrillon and A.A. Petrenko). F. Cyr received a funding from the Mourou/Strickland mobility program to work on this project. R. Tzortzis is financed by a MENRT PhD grant (École Doctorale Sciences de l'environnement - ED 251, Aix-Marseille University).

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

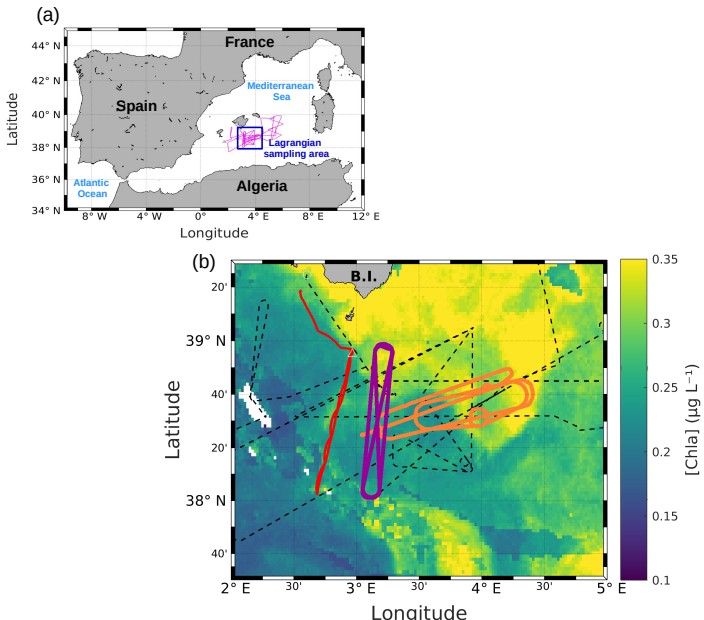

**Figure 1.** (a) Whole route of the RV *Beautemps-Beaupré* during PROTEVSMED-SWOT (pink line). The blue box corresponds to the area sampled with the Lagrangian strategy. (b) Map of satellite-derived [Chla] provided by CLS for 3 May 2018, selected in the Lagrangian sampling area and superimposed on the route of the ship (black dotted line). The orange and purple lines delimit the two areas called "hippodromes": West-East (orange) and North-South (purple). The red line represents the route of the SeaExplorer glider.

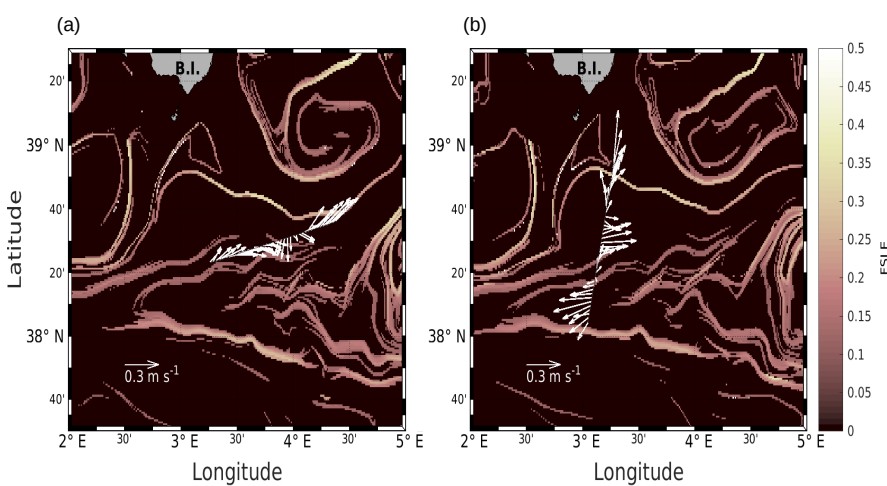

**Figure 2.** Horizontal velocities measured by VMADCP at 25 m, along the WE (a) and NS (b) transects, superimposed to the FSLE field for the corresponding date (i.e. 9 May 2018 and 11 May 2018, respectively). Unit for FSLE is $\mathrm{day}^{-1}$.

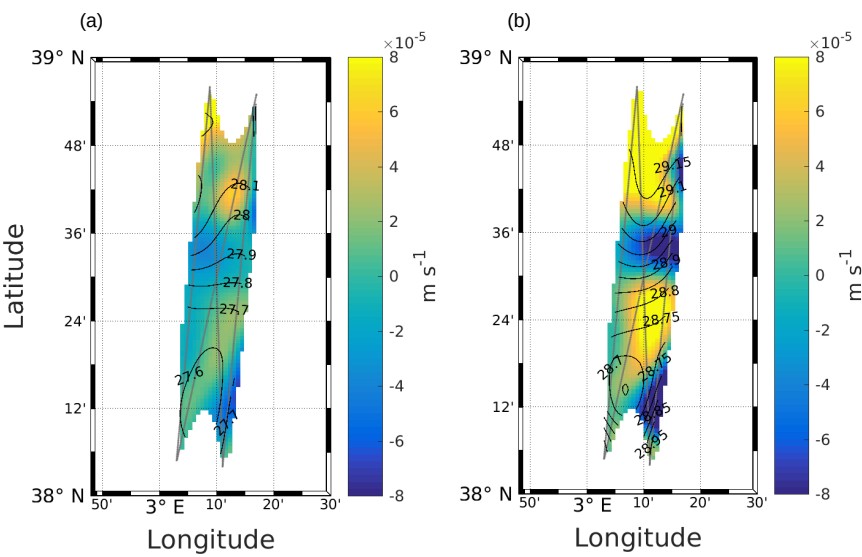

**Figure 3.** Maps of vertical velocities at (a) 25 m and (b) 85 m. Black lines represent the $\sigma$ contours. The data have been selected where the error on the objective mapping of $\sigma$ is $\leq 0.0025$.

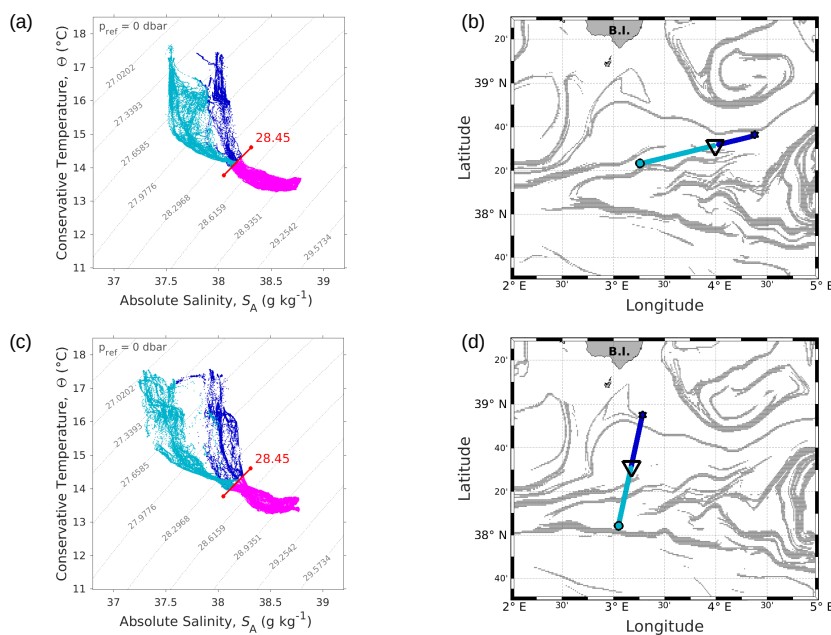

**Figure 4.** $\Theta$–$S_A$ diagrams of data collected along the WE (a) and the NS (c) transects. The "younger" AW is represented in light blue whereas the "older" AW is represented in dark blue. Intermediate water is also represented in pink. The isopycnal $28.45\ \mathrm{kg\,m^{-3}}$, separating surface waters from the deeper ones, is pointed out in red. Triangles in panels (b) and (d) indicate the geographical position of the separation between the two types of AW along the corresponding transect.

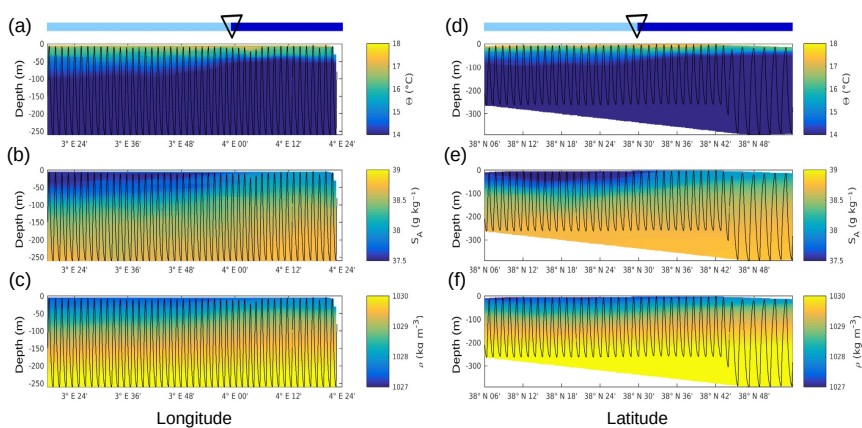

**Figure 5.** Vertical sections of conservative temperature $\Theta$ (a, d), absolute salinity $S_A$ (b, e), and density $\rho$ (c, f), sampled by the Seasoar along the WE transect (at left) and along the NS transect (at right). The Seasoar trajectory is represented by the black lines. Triangle indicates the localisation of the front area between the two types of AW represented in light and dark blue (see respectively Fig. 4b and Fig. 4d).

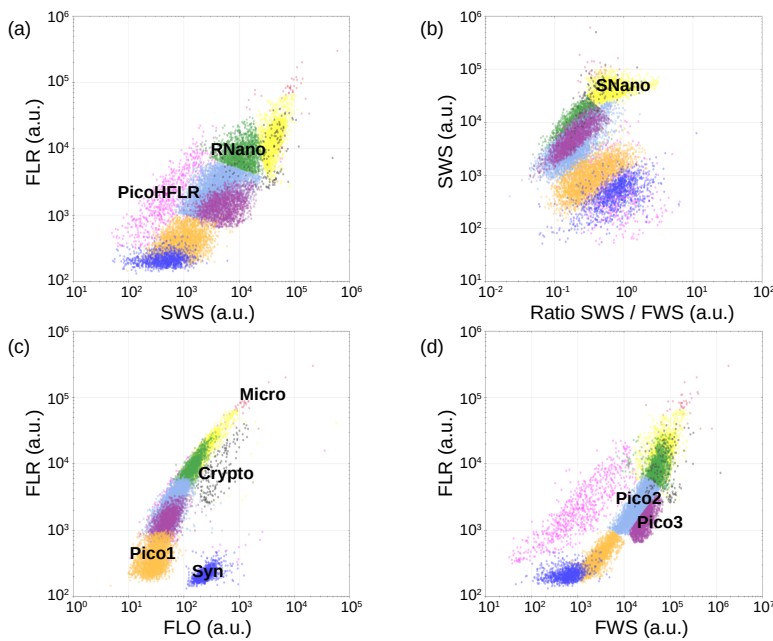

**Figure 6.** Cytograms obtained with the CytoSense automated flow cytometer. *Synechococcus* are in dark blue (Syn), the picophytoplankton with lowest FLO in orange (Pico1), the picophytoplankton with intermediate FWS in light blue (Pico2), the picophytoplankton with highest FWS in violet (Pico3), the picophytoplankton with a high red fluorescence in pink (PicoHFLR), the nanophytoplankton with high SWS/FWS ratio in yellow (SNano) and higher SWS intensities than the other nanophytoplankton (RNano) in green, the Cryptophytes in grey (Crypto) and the microphytoplankton in red (Micro). The flow cytometry units for both fluorescence and light scatter are arbitrary (a.u.).

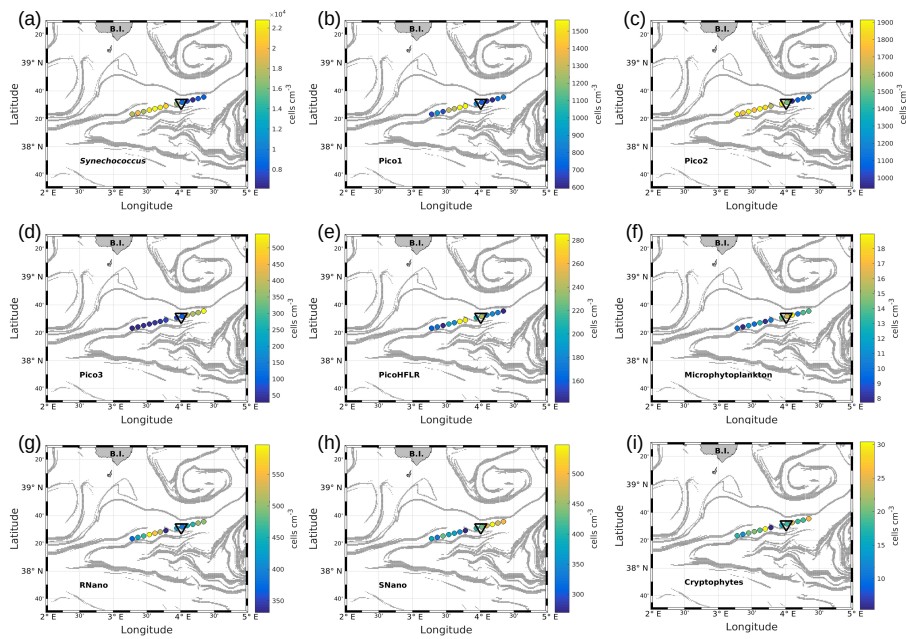

**Figure 7.** Abundances (in cells per cubic centimeter) of the phytoplankton groups along the WE transect, superimposed with the FSLE field. Triangles indicate the front area. (a) *Synechococcus* , (b) Pico1, (c) Pico2, (d) Pico3, (e) PicoHFLR, (f) Microphytoplankton, (g) RNano, (h) SNano, (i) Cryptophytes.

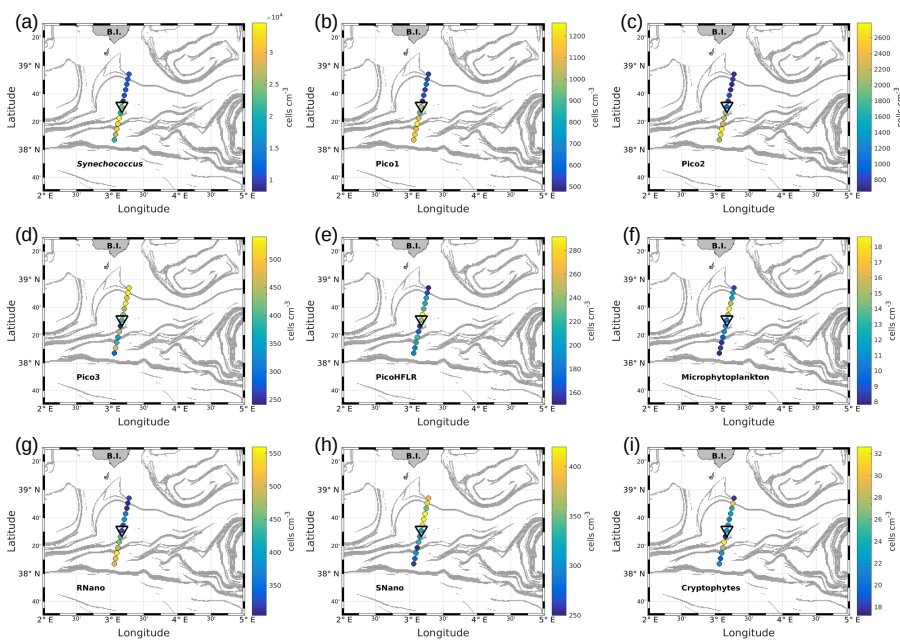

**Figure 8.** As in Fig. 7, but for NS transect.

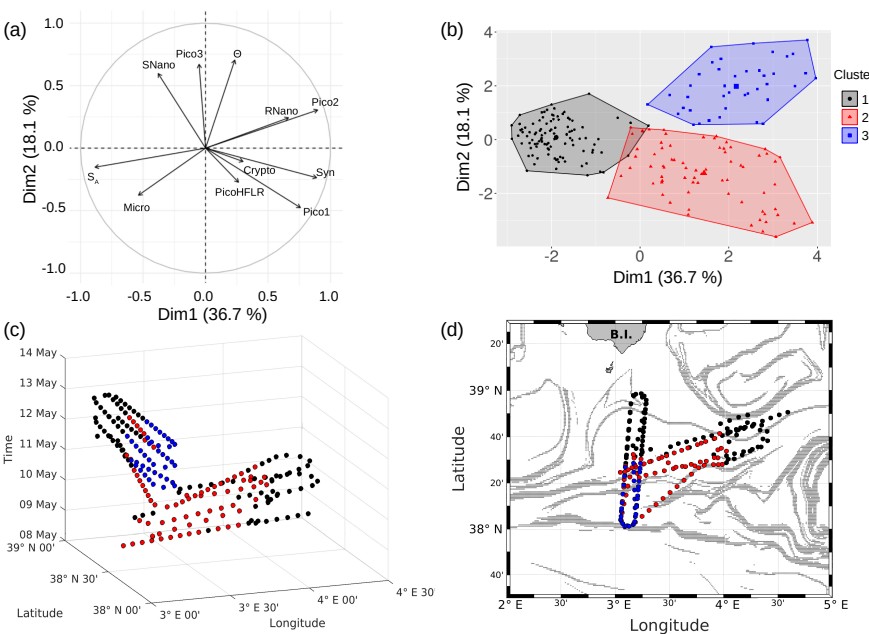

**Figure 9.** (a) Principal component analysis in the first factorial plane. (b) Representation in two dimensions of the three clusters obtained with the K-medoid algorithm. (c) Spatio-temporal representation of the three clusters obtained with the K-medoid algorithm. (d) Geographical representation of these clusters, superimposed to the FSLE field of the 11 May 2018.

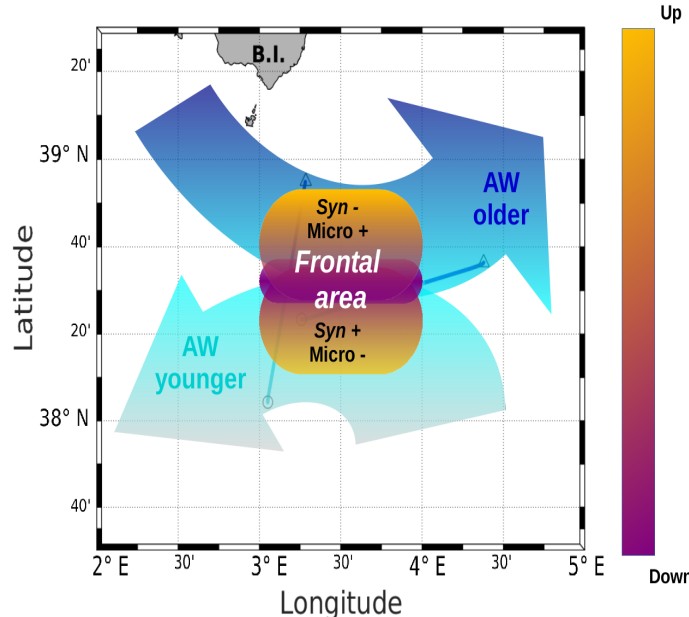

**Figure 10.** A schematic view of the of PROTEVSMED-SWOT results in the surface layer. A narrow frontal area, characterized by the change in direction of the horizontal current and by opposite vertical movements, corresponds to a rapid shift in hydrological properties and biological content of the two water masses separated by the front. For sake of simplicity only two groups of phytoplankton, *Synechococcus* (Syn) and Microphytoplankton (Micro), are indicated, but their contrasted abundances are also representative of the ones of other phytoplankton groups.

**Table 1.** Results of the iterative algorithm to determine the best separation between the two types of AW along the Seasoar transects of the WE and NS hippodromes. The lines in bold correspond to the WE and the NS transects represented in Fig. 4. Only one transect is available for WE hippodrome due to technical problems with the towed fish.

| Date of transects (Time UTC) | $(\overline{S_{A1}}, \overline{\Theta_1})$ | $(\overline{S_{A2}}, \overline{\Theta_2})$ | Max of $|\overline{S_{A1}} - \overline{S_{A2}}|$ (g kg$^{-1}$) | Lon / Lat of best separation |
|---|---|---|---|---|
| **9 May 16:50–9 May 23:45** | **(37.71, 15.62)** | **(38.02, 15.93)** | **0.31** | **4° E 06'** |
| **11 May 02:00–11 May 08:40** | **(37.63, 15.68)** | **(38.03, 15.91)** | **0.40** | **38° N 32'** |
| 11 May 10:00–11 May 16:45 | (37.59, 15.95) | (38.03, 15.69) | 0.44 | 38° N 30' |
| 11 May 17:55–12 May 00:50 | (37.63, 16.02) | (38.07, 15.61) | 0.44 | 38° N 34' |
| 12 May 01:50–12 May 08:20 | (37.63, 15.82) | (38.01, 16.15) | 0.38 | 38° N 31' |
| 12 May 09:30–12 May 16:40 | (37.63, 15.54) | (38.03, 16.09) | 0.40 | 38° N 29' |
| 12 May 17:30–13 May 00:20 | (37.75, 15.43) | (38.05, 16.30) | 0.30 | 38° N 34' |

**Table 2.** Size and abundances of the nine phytoplankton groups, identified by flow cytometry analysis on Fig. 6.

| Groups | Size (µm) | Abundances (cells cm$^{-3}$) |
| --- | --- | --- |
| *Synechococcus* | 1 | $10^4$ |
| Pico1 | 0.2–2 | $10^3$ |
| Pico2 | 0.2–2 | $10^3$ |
| Pico3 | 0.2–2 | $10^2$ |
| PicoHFLR | 0.2–2 | $10^2$ |
| RNano | 2–20 | $10^2$ |
| SNano | 2–20 | $10^2$ |
| Microphytoplankton | 20–200 | 10 |
| Cryptophytes | 10–50 | 1 |

**Table 3.** Description of the three clusters obtained with the K-medoid algorithm and represented on Fig. 9b, c and d. The local average (i.e. the average of each variables in a cluster) have been compared with the global average (i.e. the average of each variables for all the dataset) to highlight the contribution of each variable to a cluster. Note that only the five most discriminating variables, determined with the standard deviation are shown for each cluster. Absolute salinitiy is measured in gram per kilogram, conservative temperature in degree Celsius and the phytoplankton abundances in cells per cubic centimeter. The bold lines represent the variables with the most contribution of each cluster (i.e. with local average > global average).

| Clusters | Variables | Local average | Global average | Local standard deviation | Global standard deviation |
|---|---|---|---|---|---|
| 1 (Black) | **Salinity ($S_A$)** | **37.65** | **37.45** | **0.115** | **0.235** |
| | **Microphytoplankton (Micro)** | **14** | **11** | **5** | **5** |
| | **Nanophytoplankton (SNano)** | **461** | **417** | **65** | **86** |
| | Temperature ($\Theta$) | 18.1 | 18.2 | 0.4 | 0.4 |
| | Nanophytoplankton (RNano) | 386 | 454 | 81 | 140 |
| 2 (Red) | **Phytoplankton (Pico1)** | **1148** | **838** | **361** | **343** |
| | *Synechococcus* **(Syn)** | **23820** | **16949** | **5049** | **7699** |
| | **Phytoplankton (Pico2)** | **1955** | **1656** | **355** | **770** |
| | Microphytoplankton (Micro) | 10 | 11 | 4 | 5 |
| | Temperature ($\Theta$) | 18 | 18.2 | 0.3 | 0.4 |
| 3 (Blue) | **Phytoplankton (Pico2)** | **2761** | **1657** | **467** | **770** |
| | **Temperature ($\Theta$)** | **18.7** | **18.2** | **0.3** | **0.4** |
| | **Nanophytoplankton (RNano)** | **593** | **454** | **186** | **140** |
| | **Phytoplankton (Pico3)** | **557** | **355** | **280** | **212** |
| | *Synechococcus* **(Syn)** | **22151** | **16949** | **3940** | **7693** |

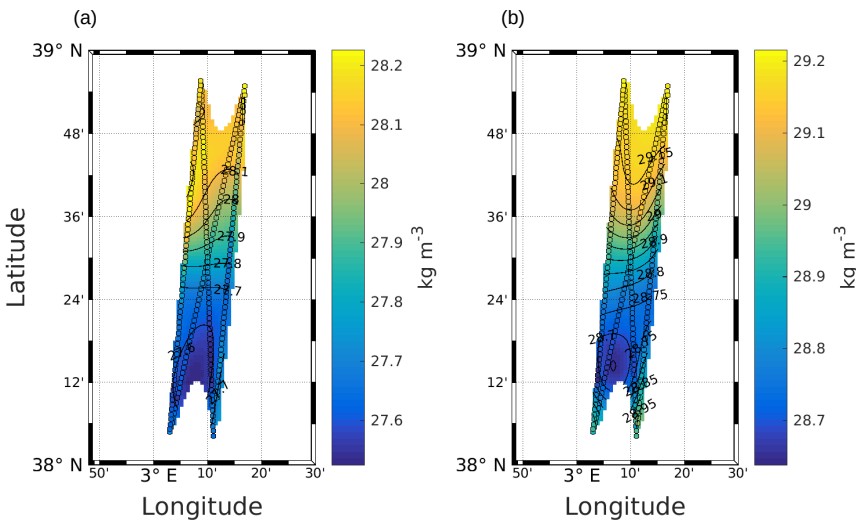

**Figure A1.** Objective mapping of potential density anomaly $\sigma$ at 25 (a) and 85 m (b). The circles represent $\sigma$ as measured by the Seasoar, along the transects used for the interpolation. Black lines represent $\sigma$ contours. The data have been selected where the error on the objective mapping of $\sigma$ is $\leq 0.0025$.

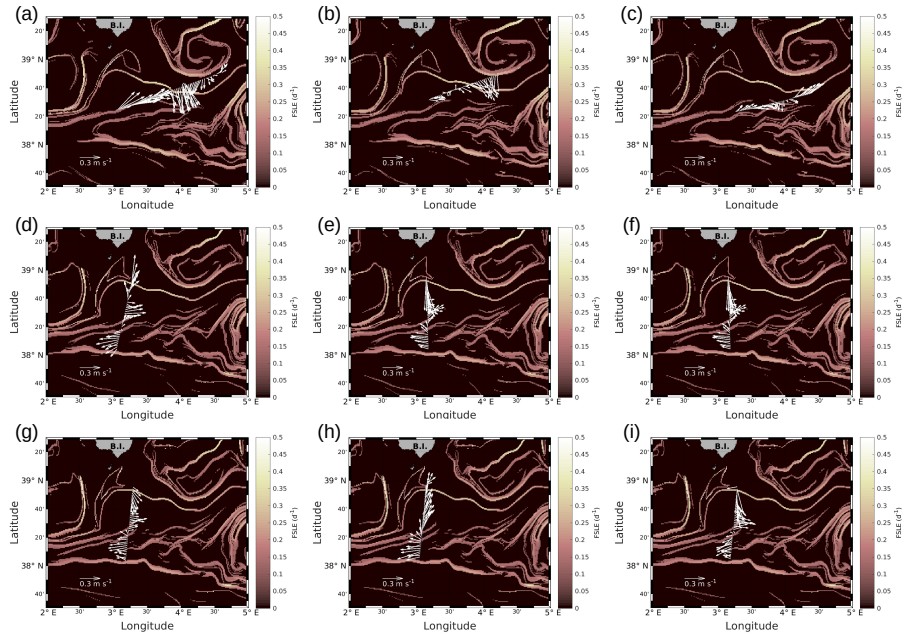

**Figure A2.** Horizontal velocities measured by VMADCP, along transects of the WE hippodrome (a), (b), (c) and the NS hippodrome (d), (e), (f), (g), (h), (i).

(a) 8 May 12:50–9 May 00:30

(b) 9 May 09:00–9 May 15:30

**(c) 9 May 16:50–9 May 23:45**

**(d) 11 May 02:00–11 May 08:40**

(e) 11 May 10:00–11 May 16:45

(f) 11 May 17:55–12 May 00:50

(g) 12 May 01:50–12 May 08:20

(h) 12 May 09:30–12 May 16:40

(i) 12 May 17:30–13 May 00:20

The lines in bold correspond to the WE and the NS transects described in the manuscript, and also represented in Fig. 4. In our study, we have chosen to select the transect (c) for the WE hippodrome, because we deplore a lack of temperature and salinity data for the other transects of the WE hippodrome, due to technical problems with the Seasoar.

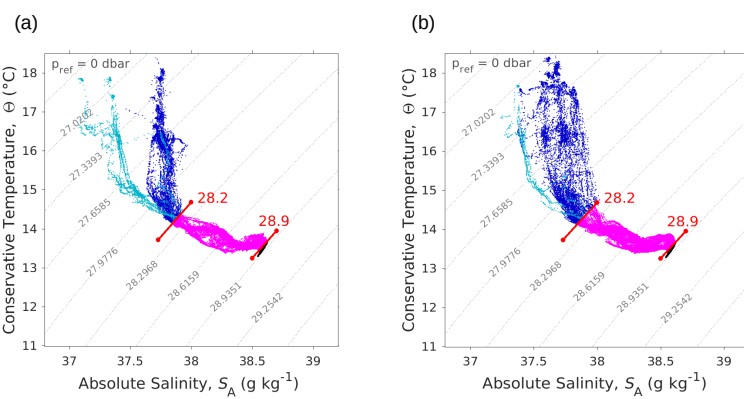

**Figure A3.** $\Theta-S_A$ diagrams measured by the SeaExplorer glider along the outward route (a): 6 May 2018 00:00–9 May 2018 21:00 UTC, and the return route (b): 10 May 2018 00:00–13 May 2018 21:00 UTC. As in Figure 4, the "younger" AW is represented in light blue whereas the "older" AW is represented in dark blue. Intermediate water and deeper water are represented in pink and black, respectively. The isopycnals 28.2 and 28.9 $\mathrm{kg\,m^{-3}}$, separating surface waters from the deeper ones, are also shown.

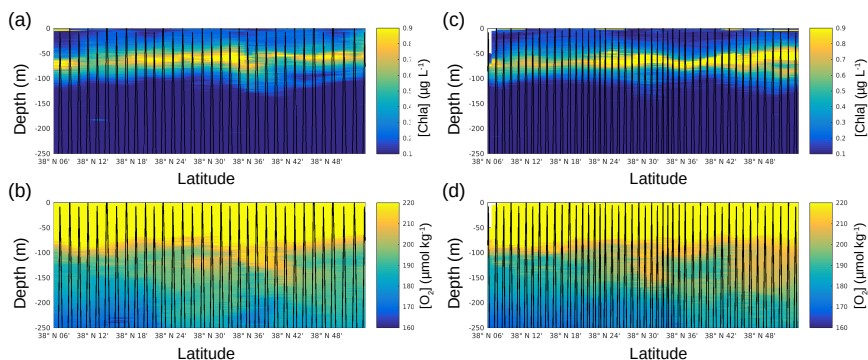

**Figure A4.** Vertical profiles of [Chla] (a, c) and dissolved oxygen concentration (b, d), measured by the SeaExplorer glider, along the outward route (at left): 6 May 2018 00:00–9 May 2018 21:00 UTC, and along the return route (at right): 10 May 2018 00:00–13 May 2018 21:00 UTC. The SeaExplorer glider trajectory is represented by the black lines. The data have been selected between the surface and 250 m for a better visualization of the surface layer.

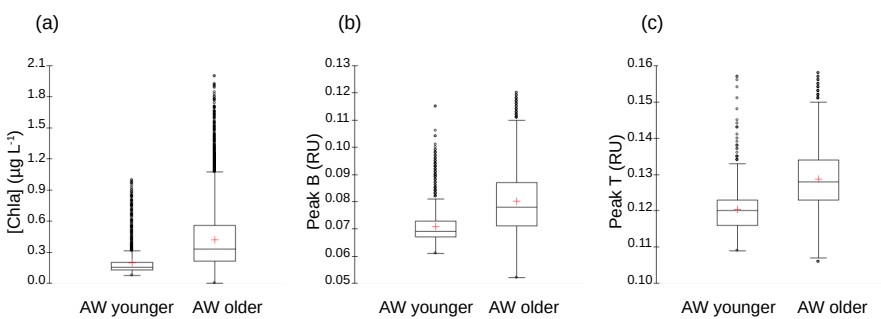

**Figure A5.** Box-and-whisker plots for the comparison of (a) [Chla] (in microgram per litre), (b) fluorescence intensities of tyrosine-like fluorophore (Peak B in RU) and (c) fluorescence intensities of tryptophan-like fluorophore (Peak T in RU) between "younger" and "older" Atlantic Waters (AW). The bottom and top of the box are the 25th and 75th percentiles, respectively, whereas the central line is the 50th percentile (the median), and the cross is the mean. The ends of the error bars correspond to the 10th percentile (bottom) and to 90th percentile (top). All the SeaExplorer glider transects are considered here (all data acquired from 6 to 15 May 2018). "Younger AW" correspond to samples showing a salinity between 37.20 and 37.84, and a temperature between 14.5 and 17.4 °C (n = 1657). "Older AW" correspond to samples displaying a salinity between 37.82 and 38.14, and a temperature between 14.0 and 18.3 °C (n = 11760). Mean values of Chla, Peak B and Peak T of "older AW" are significantly higher than those of "younger AW" (t test, p inf 0.0001).