# Peer review of "Impact of moderately energetic fine-scale dynamics on the phytoplankton community structure in the western Mediterranean Sea"

_Biogeosciences, 2021_

## Referee Comment (RC1)

**Review for Tzortzis et al. (2021)**

General comments:

The manuscript of Tzortzis et al. constitutes an interesting analysis on phytoplankton community dynamics in response to a frontal region in the Western Mediterranean Sea. Extensive *in situ* datasets are used to characterise the hydrodynamics of the region, and provide insights on the response of phytoplankton community structure to fine-scale ocean dynamics associated with the frontal region. Generally, the results are fairly well presented and are interpreted appropriately in the discussion and conclusions. One of my major issues is that the manuscript contains superfluous information/analysis at times. There are many figures and different types of analysis presented, but the authors do not summarise all of these findings in a succinct and logical manner in the discussion. In some cases, the text/analysis can be condensed and moved as supplementary material, removed, or expanded upon. Finally, although the grammar is generally OK, I would recommend it for a check by an English editing service if possible. I recommend the manuscript for major revisions prior to publication in Biogeosciences.

*Specific comments:*

Abstract

Line 7: Another word to replace "towed fish"? I presume you mean the SeaSoar?
Line 7: I think you can rephrase the sentence to: "Multi-parametric in situ sensors mounted on the vessel, a towed fish/SeaSoar instrument and an ocean glider"
Line 8: Remove "A" before "particular attention"
Line 14: "Phytoplankton community structure"?

Introduction

Line 17: rephrase to "oceanic ecosystems" and remove "the".
Line 18: The word "compartment". Perhaps it is possible to find another alternative here?
Line 19: Global *climate* change?
Line 19: Since several years? Satellites have been acquiring observations of ocean colour/phytoplankton biomass for at least 2/3 decades. I would rephrase here. Also, I think you can remove "of phytoplankton"
Line 22: I think the line following "the term fine scales" needs the addition of commas e.g., The term "fine scales" refers here to ocean dynamical processes that occur on horizontal scales of the order of 1–100 km, are characterized by a small Rossby number, and have a relatively short lifetime from days to weeks".
Line 25. "Fine scale" should be "fine scale features"?
Line 27. See previous comment
Line 31. "In respect to bulk production". I think this can be removed, as you begin a new sentence talking about diversity as opposed to primary production…
Line 32. I would replace "Indeed", with "however".
Line 32. "effect of the fine scales" – not grammatically correct. You mean "effect of fine

scale oceanic features"? Please check and fix this throughout the manuscript.

Line 34. I don't think "in situ samplings" is grammatically correct. Please check and modify if necessary.

Line 37. "associated to" should be "associated with"

Line 38. Comma needed after "surface ocean"

Materials and Methods

- If I am not mistaken, you selected the two sampling trajectories based on two regions of Chl-a concentration using the satellite-based SPASSO tool. Based on Figure 1, I can see a region of high Chl-a corresponding to the WE transect, but have trouble distinguishing the second region of unique surface Chl-a that justifies the position of the NS transect. Perhaps it is the colour scale/colour bar limits. Are the regions also selected based on SST and currents? In any case, I would rephrase or try and be more specific of why these two sampling transects were selected, and which areas you are referring to. I believe the colour scale can be improved to highlight this.

- I am not really familiar with FSLEs or Langrangian techniques. Thus, out of curiosity, is the FSLE a commonly used index for detecting fronts/fine scale features? Can you provide citations supporting this? Do maps of altimetry/SST also show the existence of the front between the two water masses?

- Line 142. "later" should be "latter"?

- Is Figure 2 absolutely necessary to include in your results? It relates mainly to your methodology and I suppose isn't overly important for the story you are trying to tell. I would consider moving this to supplementary material.

- I think it would be helpful to modify your figure 1 to show broader study region/familiar landmarks, so readers not familiar with the Mediterranean Sea can get more of an idea of the region you are working in.

Results

- It would help to try and highlight the specific zonal feature being discussed in Figure 3. I can see several features based on the FSLE map, corresponding to the latitude 38$^o$ N 20$^'$.

- Out of curiosity, do the other transects (not presented) show the same results?

- Line 218. I would help the reader and refer to your figures here (Figs. 5b and d). I suppose the triangles indicate the position of this?

- Line 225 onwards. This is your results section and thus I would avoid trying to discuss your observations using citations. Perhaps this information can be moved to the discussion.

- Lines 237-239. Again, this seems more like discussion material.
Furthermore, although it is nice that you have shown similar results in temperature and salinity using an independent glider dataset, is the addition of a figure necessary here? You can probably briefly mention that the glider dataset showed similar results. I only mention this as the manuscript text is relatively short, and yet you have 16 figures. I would think about condensing your analysis slightly and think about where figures may be more appropriate as supplementary material.

- Following my previous comment, Figures 7 and 8 are not described in much detail. For example, Lines 241 – 244 are fairly broad, considering you are talking about three separate transects, for each hippodrome in Figures 7 and 8. I would try and be more clear and descriptive with your results here. Indeed, the data show a clear, interesting separation between the two different water masses (although note that this is less apparent in your density plots…).

- Line 247. For your DO and Chl-a plots, please re-clarify what hippodrome you are referring to (the NS one).

- Lines 247 – 249. What does "richness of structures" mean? Please be more descriptive with your results, or otherwise, remove superfluous material.

- Line 251. Chla is higher where exactly? Please expand and provide detail to your analysis.

- Lines 251 – 254. What plots are you referring to here? Also avoid general explanations in your results, especially without providing evidence or context e.g. "probably associate with vertical dynamics of the front". Provide more details (in the discussion) or remove. Please go through the whole manuscript and avoid such general statements.

- Line 255 onwards. I am struggling to understand the connection you are trying to make between Chl-a/DO and your "peak T/peak B". In the methodology, it is not fully clear to me what the motivation was for measuring these parameters. Please clarify. Furthermore, you do not really discuss these parameters in your discussion. Please consider what information is directly relevant to your analysis and justify the inclusion of each of your figures with corresponding text.

Discussion/conclusions

- Line 361. I would avoid using informal text like "thanks to the flow cytometry measurements".

- Lines 349 -359. Why don't you mention the consequences of these dynamics in terms of upwelling/downwelling here? This is what is driving your phytoplankton variability after all?

-  379 -380. Please expand on this! How does all of your statistical analysis support your results? And why is there no reference to the figures highlighting this analysis?

- Lines 337 – 348. This is quite confusing as written.

- Lines 362 onwards. What about the other phytoplankton groups identified by flow cytometry? You simplify here that there is two main groups, and yet quite an in depth analysis is presented for the other groups in figures 12 -15.

-  What contributes to the variability in phytoplankton community structure along the WE hippodrome? It is quite clear you have two distinct northern and southern water masses, but I wonder if there are other physical mechanisms that may be driving the variability you see longitudinally? What about the horizontal movement of water masses?

- Overall, the discussion needs to fully encapsulate the results that you present. As it stands currently, it appears at times to only take bits and pieces of your story and I feel much of your previous analysis is ignored.

---

## Author Comment (AC1)

Dear referee,

Thank you very much for your constructive comments and suggestions, as well as your English corrections.

**General comments**

You have highlighted that the major problem in the manuscript is that it contains superfluous information, in particular in the Results and in the Discussion. We will modify these parts taking into account your suggestions. The second referee has also sent his comments. Following the Biogeosciences guidelines, we have first to provide you a feedback about the main points you raised, and then, only after the Editor's green light, we will modify the manuscript. In order to follow our reply, your comments have been copied here after the '==>' symbol.

**Specific comments**

**Abstract and Introduction**

Thank you for your English corrections, as mentioned above, we will rework the manuscript after the answer of Biogeosciences.

**Materiels and Methods**

==> "If I am not mistaken, you selected the two sampling trajectories based on two regions of Chl-a concentration using the satellite-based SPASSO tool. Based on Figure 1, I can see a region of high Chl-a corresponding to the WE transect, but have trouble distinguishing the second region of unique surface Chl-a that justifies the position of the NS transect. Perhaps it is the colour scale/colorbar limits. Are the regions also selected based on SST and currents? In any case, I would rephrase or try and be more specific of why these two sampling transects were selected, and which areas you are referring to. I believe the colour scale can be improved to highlight this."

SPASSO was used to follow both the temporal and spatial variability of the horizontal finescale features of interest. SPASSO combines satellite-derived currents, SST and [Chla], to provide maps of dynamical and biogeochemical structures in both Near Real Time (NRT) and Delayed Time (DT). For simplicity, we have chosen to show only a map of [Chla] in Fig. 1. During the cruise, the analysis of maps provided by SPASSO suggested the presence of two different regions characterized by their different surface [Chla]. The complete archive of figures is available at https://spasso.mio.osupytheas.fr/PREBIOSWOT/Figures\_web/. A new figure is proposed below with modified color ranges in the hope that the gradients of [Chla] are better put in evidence.

(a) Route of the RV Beautemps-Beaupré during PROTEVSMED-SWOT (pink line). The blue box corresponds to the area sampled with Lagrangian strategy. (b) Map of satellite-derived [Chla] provided by CLS for 3 May 2018, selected in the Lagrangian area and superimposed on the route of the ship (black dotted line). The orange and purple lines delimit the two areas called "hippodromes": West-East (orange) and North-South (purple). The red line represents the route of the SeaExplorer glider.

The high [Chla] region of interest, i.e., with a concentration greater than 0.3  $\mu$ g L-1, is located between 3° E and 4° E 30', with its southern latitude varying from 38° N 20' for longitudes between 4° E and 4° E 30' to 38° N 40' for longitudes around 3° E. The Lagrangian strategy consisted in sampling longitudinally and latitudinally the high and low regions of [Chla]. The dedicated route of the ship across these two regions is represented in purple and in orange on Fig. 1. A special attention was paid to adapt the temporal sampling in these different water masses to the biological time scales, i.e. trying to catch the diurnal cycle. Because of the shape of the ship track, in the following we refer to these areas as "West-East (WE) hippodrome" (in orange on Fig. 1) performed from 8 May 15:30 to 10 May 17:30 UTC, and "North-South (NS) hippodrome" (in purple on Fig. 1) performed between 11 May 02:00 and 13 May 08:30 UTC.

**==> "I am not really familiar with FSLEs or Lagrangian techniques. Thus, out of curiosity, is the FSLE a commonly used index for detecting fronts/fine scale features? Can you provide citations supporting this?"**

The first study that showed the interest of using FSLE-derived fronts for biogeochemical studies was probably Lehahn et al., 2007 (see in particular their Fig. 8 and 9). Before, Abraham and Bowen (2002) have been the first to apply the Lyapunov exponent technique (although finite-time, not finite-size) to the ocean, in turn borrowing some ideas from dynamical system theory (see in particular Boffetta et al., 2001). For campaign studies, the FSLE analysis permits to identify biogeochemical regions of potential interest. This strategy has already been tested, either in post-cruise or real-time analysis, during many campaigns such as LOHAFEX (Smetacek et al., 2012), Latex10 2010 (Petrenko 2010), KEOPS2 (d'Ovidio et al., 2015), STRASSE 2012, OUTPACE 2015 (Rousselet et al., 2018; de Verneil et al., 2019), OSCAHR 2015 (Marrec et al., 2018; Rousselet et al., 2019), PEACETIME 2017, SARGASSES 2017, FUMSECK 2019 (Barrillon 2019; Comby et al., 2021), TONGA 2019 (Benavides et al., 2021) and SWINGS 2021 to identify structures of interest. A review on the FSLE and other satellite-based Lagrangian techniques can be found in Lehahn et al., 2018).

**==> "Do maps of altimetry/SST also show the existence of the front between the two water masses?"**

In our work altimetry is the input data for the FLSE (see above). At 38° N 20', the altimetryderived surface current directions change drastically along the NS transect, suggesting a front. However, this is not clearly the case for the WE transect. We don't think altimetric maps are essential for the paper because the front is clearly visible with the VM-ADCP current and the FSLE (see figures above, or eventually in the future supplementary material). Maps of SST can provide another view of fine-scale dynamics. However, the front is not clearly visible on the map of SST. Indeed, gradients of temperature are not enough contrasted in spring, in the Mediterranean Sea. It is easier to locate the front with the map of [Chla] (cf Fig. 1) than the SST, that is why we think these maps aren't necessary for the paper.

---

## Author Comment (AC2)

Dear referee,

Thank you for the attention that you have given to our work.

We provide below, a feedback about the main points you raised. We are confident that following your remarks and suggestions, as well as the ones from the other anonymous referee, we can improve our manuscript for publication in Biogeosciences. In order to follow our reply, your comments have been copied hereafter the '==>' symbol.

**General comments**

==> *"The impact of fine-scale physical processes on plankton community is indeed very important. We have fully realized the significance of this problem, but limited by the observation means (especially biological parameters), the current understanding is very limited. Based on these backgrounds, I think this work is a very good attempt. The author's cruise design is very targeted, various equipment is very effective, the text description is very clear and detailed. However, from the perspective of research papers, I did not see the logical chain driven by scientific hypotheses. Instead, they used various devices to verify some predictable results. If the physical-biological processes and mechanisms in fine-scale are consistent with those in meso-/large- scales, why are they so important and are unique? When the scale becomes smaller, what is the most important scientific question in the process of physical-biological processes? Because this part is not highlighting enough, I have been looking forward to the new results (differences with large-scale and classical observation) and thinking about so what? What is the implication? As a research paper, I would like to see the author's point around a new result, or a logical inference."*

First of all, we kindly disagree with the Reviewer on the fact that showing consistency between some biophysical stirring processes in meso- and large- scales is a non-result or a result of poor interest. Such a consistency should not be automatically expected. An equally plausible scenario could have been that, in regions that are much less energetic than boundary currents, like the region we studied, the physico-chemical contrasts induced by the horizontal stirring are not sufficiently strong to spatially reflect into different phytoplanktonic communities. That said, we acknowledge that we did not detail our scientific questioning enough and we were not clear enough about the novelty and the implications of our results. Moreover, we did not sufficiently discuss how the dynamical characteristics of the region specifically structure the phytoplankton community. As mentioned in the Introduction, horizontal fine-scales have been predominantly studied with numerical simulations and satellite observations. The validation of these studies is however difficult due to the lack of in situ observations, especially biological parameters. It is indeed a real challenge to perform in situ measurements of fine-scales processes due to their small physical scales and short lifetimes. In recent years only few in situ samplings targeting the fine scales have been performed, and the important point is that most of these cruises have been led in large and

energetic regions such as in coastal upwelling regions (Ribalet et al., 2010) and in boundary currents (Clayton et al., 2014 ; 2017) known to generate persistent fronts and to host biodiversity hotspot areas (Barton et al., 2014 ; Lévy et al., 2015). Contrary to the existing and growing body of literature, this study is one of the only few targeting particularly less energetic and more ephemeral fine-scale processes (e.g., typical of Mediterranean Sea) and their effects on phytoplankton. Our results show a barrier role played by a front on the distribution of phytoplankton abundances. The fact that a moderately energetic fine scale front plays a similar role as large fronts in structuring the phytoplankton community has never been shown before and consists of the key original result of this study. Moreover, contrary to previous studies performed in nutrient-rich areas, our results have been obtained in an oligotrophic region. Since oligotrophic and weaker energetic regions are more representative of the global ocean than highly energetic structures presented in previous works, we are convinced that our results shed light on a better understanding of the functioning of the global ocean, and are therefore of great interest to the readers of Biogeosciences. We plan to stress these findings in the revised version of our manuscript.

*==> "I noticed that there are two different water masses, one is old AW and the other is young AW. The later data analysis is almost organized according to this logic. Although the traditional physical ocean observation (such as water mass analysis) can also distinguish these two water masses generally, I found that the biological parameters do not seem to be completely consistent (at least some mathematical analysis is needed to clarify from the seemingly chaotic distribution). If the author can dig in depth according to this logic and see if the underlying mechanism is universal (extrapolation), they may be able to find a clue."*

The clustering analysis of flow cytometry data allowed to detect several groups of phytoplankton (cf Fig. 12 in the manuscript), and, in particular, to identify various groups of eukaryotic Nanophytoplankton (RNano and SNano) and eukaryotic Picophytoplankton (Pico1, Pico2, Pico3, and PicoHFLR). Nevertheless, in the first version of the manuscript, we showed figures where the abundances of nanophytoplankton represented the sum of the abundances of RNano and SNano groups. Thanks to your remarks, we have redone Fig. 13 and 14 (cf figures below) taking into account RNano and SNano abundances separately.

[Figure]

Abundances (in cells per cubic centimeter) of the phytoplankton groups along the WE transect, superimposed with the FSLE field. Triangles indicate the front area.

[Figure]

Abundances (in cells per cubic centimeter) of the phytoplankton groups along the NS transect, superimposed with the FSLE field. Triangles indicate the front area.

On these new figures, it is possible to distinguish a clear separation of RNano and SNano abundances by the front. Furthermore, the distribution of RNano abundance is opposite to the SNano abundance. This opposite distribution is very likely the reason for the unclear distribution that you noticed when both groups were merged. For the sake of consistency, the pico-phytoplankton groups (Pico1, Pico2, Pico3 and PicoHFLR) will be separated in the new version of the manuscript, which wasn't the case in the first version. In the new figure above, Pico1, Pico2, and Pico3 abundances appear now clearly separated by the front. The distribution of PicoHFLR, as well as Cryptophytes, remains less clear. These groups do not appear as well correlated with temperature and salinity (used for the characterization of water masses), unlike the other phytoplankton groups. One explanation could be that these cells could be less sensitive to the environmental conditions than the other groups. Another explanation could be their low abundances combined to the fact that these groups are more difficult to define than the other: the limit of the groups is less obvious and maybe some of the events gated as PicoHFLR or Cryptophytes may be in fact some background noise.

The principal component analysis (Fig. 15a) and the K-medoid algorithm (Fig. 15b) already constitute an advanced mathematical analysis. The PCA results (cf Fig. 15a in the manuscript) clearly indicate what is qualitatively observed in the new figures and described above: the opposite distribution of the RNano and SNano abundance, the front separation for

the picophytoplankton abundances.

==> *"Although the author defines fine scale (line 23), there are other related descriptions, which are easy to be confused. For example, low energetic front (lines 53, 60) and moderate energetic front (Title. Line 5, 328)."*

Thank you. We will harmonize the text in order to use only the term "moderate energetic front".

==> *"The size of phytoplankton is also confused. In fact, Synechococcus belongs to picophytoplankton, and I guess the "picophytoplankton" in the article means eukaryotic picophytoplankton. In addition, most cryptophytes are considered to be in nano size. Anyway, there's some confusion."*

As mentioned above, we have identified several groups of phytoplankton by flow cytometry and used the conventional names used by flow cytometrists. Phytoplankton groups were resolved on the basis of their light scatter (namely forward scatter FWS and sideward scatter SWS) and fluorescence (red FLR and orange FLO fluorescence ranges) properties (Thyssen et al., 2015 ; Marrec et al., 2018). The name are indeed confusing in the manuscript as some groups are related to the taxonomy (*Synechococcus*, Cryptophytes) while the others are more related to a range of size (picoeukaryotes, nanoeukaryotes). We will correct that in the new version, to help the reader. Indeed, for instance, *Synechococcus* belongs to picophytoplankton, but it is a prokaryote. We have decided to keep it separated in the study because of that and because it was unambiguously put in evidence by flow cytometry thanks to its higher FLO intensity induced by the presence of phycoerythrin pigments. Idem for Cryptophytes which can be pico- or nanoeukaryotes but can also be discriminated from the red-only fluorescing pico- or nanoeukaryotes based on their orange fluorescence.

**Minor comments and suggestions**

==> *"Abstract: It is necessary to present some new results based on fine scale observations."*

==> *"Line 259, n=≈?"*

==> *"Line 323-331, It is repeated in the introduction"*

==> *"Line 365-370: I suggest more analysis, based on your high-resolution results, to give more evidence or new explanations."*

We will rework the manuscript to take into account your useful suggestions.

==> *"Line 373, The concentration and structure of dissolved organic matter may be controlled by physical processes (aggregation and dilution) on the one hand, and by biological effects (production rate and species composition) on the other. Whether these high-resolution matching data can be further mined."*

Thank you for your suggestion concerning the interpretation of the structure of dissolved organic matter.

Tryptophan- and tyrosine-like FDOM fluorophores (peaks T and B, respectively) are recognized to have an autochthonous origin in the marine environment, being produced through the activity of autotrophic and heterotrophic plankton organisms, in particular phytoplankton and heterotrophic bacteria (Stemond and Cory, 2014), and are known to be indicators of bioavailable/labile DOM (C and N) (Hudson et al., 2008; Fellman et al., 2009). Even though phytoplankton activity is considered a source of tryptophan- and tyrosine-like fluorophores (Determann et al., 1998; Stedmon and Markager, 2005; Romero-Castillo et al., 2010), bacterial degradation appears to be a source, but also a sink for these fluorophores, depending on the availability in nutrients (Cammack et al., 2004; Nieto-Cid et al., 2006; Biers et al., 2007).

In the present work, higher contents in tryptophan- and tyrosine-like fluorophores were found in the northern part of the transect ("older" AW) relative to the southern part ("young" AW). The same distribution pattern was observed for total Chla and O2 concentrations, as well as microphytoplankton abundance. These results highlight the strong coupling between hydrology, phytoplankton activity and DOM concentration in this area. In addition, it has been recently shown that various groups of microphytoplankton might produce tryptophan- and tyrosine-like fluorophores (Romero-Castillo et al., 2010; Fukuzaki et al., 2014; Retelletti Brogi et al., 2020), which is in agreement with our observations. The fact that tyrosine-like fluorophore was rather associated with Chla concentration and tryptophan-like with O2 concentration reveals that these two fluorophores were probably not issued from the same phytoplankton groups. Moreover, it seems that tryptophan would be more susceptible to be released by heterotrophic bacteria (in addition to be released by phytoplankton) than would be tyrosine-like material (Hudson et al., 2008; Tedetti et al., 2012; Stemond and Cory, 2014).

==> *"Line 383, "provide an in-situ confirmation of the findings" may not enough. In fact, your data is so good that you don't need to prove other people's opinions at all."*

==> *"Line 387, change "Kurushio" to "Kuroshio"."*

As mentioned above, we will rework the manuscript after the answer of Biogeosciences.

==> *"Figure 1, large scale circulation and map may help readers understand better."*

Concerning Figure 1, we have done a new figure below with a larger area.

[Figure]

(a) Route of the RV Beautemps-Beaupré during PROTEVSMED-SWOT (pink line). The blue box corresponds to the area sampled with Lagrangian strategy. (b) Map of satellite-derived [Chla] provided by CLS for 3 May 2018, selected in the Lagrangian area and superimposed on the route of the ship (black dotted line). The orange and purple lines delimit the two areas called "hippodromes": West-East (orange) and North-South (purple). The red line represents the route of the SeaExplorer glider.

==> *"Some figures can be combined and are more suitable for comparison, as Figures 2 and 4, Figures 3 and 5, and Figures 6-10."*

We will also combine figures and, following the suggestion of the other referee, some figures could be moved in supplementary material.

**References**

Barton, A. D., Ward, B. A., Williams, R. G., and Follows, M. J.: The impact of fine-scale turbulence on phytoplankton community structure, Limnology and oceanography: fluids and environments, 4, 34–49, https://doi.org/10.1215/21573689-2651533, 2014.

Biers, E.J., Zepp, R.G., Moran, M.A.: The role of nitrogen in chromophoric and fluorescent dissolved organic matter formation. Marine Chemistry, 103, 46–60. https://doi.org/10.1016/j.marchem.2006.06.003, 2007.

Cammack, W.K., Kalff, J., Prairie, Y.T., Smith, E.M.: Fluorescent dissolved organic matter in lakes: Relationships with heterotrophic metabolism. Limnology and Oceanography, 49, 2034–2045. https://doi.org/10.4319/lo.2004.49.6.2034, 2004.

Clayton, S., Nagai, T., and Follows, M. J.: Fine scale phytoplankton community structure across the Kuroshio Front, J. Plankton. Res., 36, 1017–1030, https://doi.org/10.1093/plankt/fbu020, 2014.

Clayton, S., Lin, Y.-C., Follows, M. J., and Worden, A. Z.: Co-existence of distinct Ostreococcus ecotypes at an oceanic front, Limnol. Oceanogr., 62, 75–88, https://doi.org/10.1002/lno.10373, 2017.

Determann, S., Lobbes, J.M., Reuter, R., Rullköter, J.: Ultraviolet fluorescence excitation and emission spectroscopy of marine algae and bacteria. Marine Chemistry, 62, 137–156. https://doi.org/10.1016/S0304-4203(98)00026-7, 1998.

Fellman, J.B., Hood, E. D'Amore, D.V., Edwards, R.T., White, D.: Seasonal changes in the chemical quality and biodegradability of dissolved organic matter exported from soils to streams in coastal temperate rainforest watersheds. Biogeochemistry, 95, 277–293. https://doi.org/10.1007/s10533-009-9336-6, 2009.

Fukuzaki, K., Imai, I., Fukushima, K., Ishii, K.-I., Sawayama, S., Yoshioka, T.: Fluorescent characteristics of dissolved organic matterproduced by bloom-forming coastal phytoplankton. Journal of Plankton Research, 36, 685-694. https://doi.org/10.1093/plankt/fbu015, 2014.

Hudson, N., Baker, A., Ward, D., Reynolds, D.M., Brunsdon, C., Carliell- Marquet, C., Browning, S.: Can fluorescence spectrometry be used as a surrogate for the Biochemical Oxygen Demand (BOD) test in water quality assessment? An example from South West England. Science of the Total Environment, 391, 149–158. https://doi.org/10.1016/j.scitotenv.2007.10.054, 2008.

Lévy, M., Jahn, O., Dutkiewicz, S., Follows, M. J., and d'Ovidio, F.: The dynamical landscape of marine phytoplankton diversity, J. Roy.Soc. Interface, 12, 20150 481, https://doi.org/10.1098/rsif.2015.0481, 2015.

Marrec, P., Grégori, G., Doglioli, A. M., Dugenne, M., Della Penna, A., Bhairy, N., Cariou, T., Hélias Nunige, S., Lahbib, S., Rougier, G., Wagener, T., and Thyssen, M.: Coupling physics and biogeochemistry thanks to high-resolution observations of the phytoplankton community structure in the northwestern Mediterranean Sea, Biogeosciences, 15, 1579–1606, https://doi.org/10.5194/bg-15-1579-2018, 2018.

Nieto-Cid, M., Alvarez-Salgado, X.A., Perez, F.F.: Microbial and photochemical reactivity of fluorescent dissolved organic matter in a coastal upwelling system. Limnology and Oceanography, 51, 1391–1400. https://doi.org/10.4319/lo.2006.51.3.1391, 2006.

Retelletti Brogi, S., Charrière, B., Gonnelli, M., Vaultier, F., Sempéré, R., Vestri, S., Santinelli, C.: Effect of UV and Visible Radiation on Optical Properties of Chromophoric Dissolved Organic Matter Released by Emiliania huxleyi. Journal of Marine Science and Engineering. 28, 888. https://doi.org/10.3390/jmse8110888, 2020.

Ribalet, F., Marchetti, A., Hubbard, K. A., Brown, K., Durkin, C. A., Morales, R., Robert, M., Swalwell, J. E., Tortell, P. D., and Armbrust, E. V.: Unveiling a phytoplankton hotspot at a narrow boundary between coastal and offshore waters, Proc. Nat. Acad. Sci. USA, 107, 16 571–16 576, https://doi.org/10.1073/pnas.1005638107, 2010.

Romera-Castillo, C., Sarmento, H., Álvarez-Salgado, X.A., Gasol, J.M., Marrase C.: Production of

chromophoric dissolved organic matter by marine phytoplankton. Limnology and Oceanography, 55, 446–454. https://doi.org/10.4319/lo.2010.55.1.0446, 2010.

Stedmon, C.A., Markager, S.: Tracing the production and degradation of autochthonous fractions of dissolved organic matter using fluorescence analysis. Limnology and Oceanography, 50, 1415–1426. https://doi.org/10.4319/lo.2005.50.5.1415, 2005.

Stedmon, C.A., Cory, R.M., 2014. Biological origins and fate of fluorescent dissolved organic matter in aquatic environments. In: Aquatic organic matter fluorescence. Edited by P.G. Coble, J. Lead, A. Baker, D.M. Reynolds and R.G.M. Spencer. Cambridge Environmental Chemistry Series, Cambridge University Press, New York, USA, pp. 278–299. ISBN: 9780521764612.

Tedetti, M., Longhitano, R., Garcia, N., Guigue, C., Ferretto, N., Goutx, M.: Fluorescence properties of dissolved organic matter in coastal Mediterranean waters influenced by a municipal sewage effluent (Bay of Marseilles, France). Environmental Chemistry, 9, 438–449. https://doi.org/10.1071/EN12081, 2012.

Thyssen, M., Alvain, S., Lefèbvre, A., Dessailly, D., Rijkeboer, M., Guiselin, N., Creach, V., and Artigas, L.-F.: High-resolution analysis of a North Sea phytoplankton community structure based on in situ flow cytometry observations and potential implication for remote sensing, Biogeosciences, 12, 4051–4066, https://doi.org/10.5194/bg-12-4051-2015 , 2015.

---

## Author Response (AR1)

Dear referee,

5

30

Thank you very much for your constructive comments and suggestions, as well as your English corrections. Biogeoscience has given its green light to revise our paper. So we have reworked the manuscript, taking into account your suggestions and those of the other reviewer. In order to follow our reply, your comments have been copied here in blue, after the '==>' symbol.

**General comments**

==> The manuscript of Tzortzis et al. constitutes an interesting analysis on phytoplankton
 community dynamics in response to a frontal region in the Western Mediterranean Sea. Extensive in situ datasets are used to characterise the hydrodynamics of the region, and provide insights on the response of phytoplankton community structure to fine-scale ocean dynamics associated with the frontal region. Generally, the results are fairly well presented and are interpreted appropriately in the discussion and conclusions. One of my major issues is

- 15 that the manuscript contains superfluous information/analysis at times. There are many figures and different types of analysis presented, but the authors do not summarise all of these findings in a succinct and logical manner in the discussion. In some cases, the text/analysis can be condensed and moved as supplementary material, removed, or expanded upon. Finally, although the grammar is generally OK, I would recommend it for a check by an
- **20** English editing service if possible. I recommend the manuscript for major revisions prior to publication in Biogeosciences.

You have highlighted that the major problem in our manuscript was some superfluous information, in particular in the Results and in the Discussion. We thank Reviewer 1 for this concern and have modified these parts taking into account your suggestions and those of the other reviewer as well in order to improve the manuscript. Indeed, in the revised version of our manuscript, we have separated the Discussion and the Conclusion, and we also have divided the Discussion into several subsections to help the reader.

In the first subsection of the Discussion (part 4.1 "Physical properties of the front"), we have developed our ideas about the estimation of vertical velocities in our case, and then we have dug deeper into our interpretation about the two AW observed and their role in the structuring of the front.

The second subsection (part 4.2 "Biogeoschemistry") resumes some points that we had mentioned in the Results section of the initially submitted manuscript. In this revised version, we have more detailed the implication of these results for our study.

35 In the third subsection (part 4.3 "Physical-biological coupling in the frontal area"), we have clarified the role played by the front physical forcing on the distribution of phytoplankton abundances.

Finally, in our "Conclusion and Perspectives" (part 5), we have added some sentences (lines 438-446) in order to clarify and highlight the novelty and the implications of our study.

Furthermore, we have combined figures, following your suggestions and those of the other 40 referee (see figure 5 in the new manuscript). Some figures have also been moved in Appendices (see figures A1, A2, A3, A4 and A5).

**Specific comments**

**Abstract**

==> Line 7: Another word to replace "towed fish"? I presume you mean the SeaSoar? 45

==> Line 7: I think you can rephrase the sentence to: "Multi-parametric in situ sensors mounted on the vessel, a towed fish/SeaSoar instrument and an ocean glider"

Yes, you are right, we deployed a SeaSoar; we prefer to avoid the use of the commercial name in the abstract (it is specified in the Method section) and we modified the sentence by using "towed vehicle", see line 7 (page 1) in the revised manuscript.

==> Line 8: Remove "A" before "particular attention"

We have corrected that, see line 8 (page 1) in the revised manuscript.

==> Line 14: "Phytoplankton community structure"?

We have modified the end of the abstract taking into account the comments of the other reviewer, in order to highlight the originality of our study. See lines 13-18 (page 1) in the 55 revised manuscript.

**Introduction**

==> Line 17: rephrase to "oceanic ecosystems" and remove "the".

==> Line 18: The word "compartment". Perhaps it is possible to find another alternative here?

60

50

==> Line 19: Global climate change?

We have modified the first sentence of the introduction, see lines 20-23 (page 2) in the revised manuscript.

==> Line 19: Since several years? Satellites have been acquiring observations of ocean colour/phytoplankton biomass for at least 2/3 decades. I would rephrase here. Also, I think 65 you can remove "of phytoplankton"

See line 24 (page 2) in the revised manuscript.

==> Line 22: I think the line following "the term fine scales" needs the addition of commas e.g., The term "fine scales" refers here to ocean dynamical processes that occur on horizontal scales of the order of 1–100 km, are characterized by a small Rossby number, and have a 70 relatively short lifetime from days to weeks".

Done, see lines 27-28 (page 2) in the revised manuscript.

==> Line 25. "Fine scale" should be "fine scale features"?

Done, see line 29 (page 2) in the revised manuscript.

75 ==> Line 27. See previous comment

Done, see line 31 (page 2) in the revised manuscript.

==> Line 31. "In respect to bulk production". I think this can be removed, as you begin a new sentence talking about diversity as opposed to primary production...

Done

**80** ==> Line 32. I would replace "Indeed", with "however".

**Done**

==> Line 32. "effect of the fine scales" – not grammatically correct. You mean "effect of fine-scale oceanic features"? Please check and fix this throughout the manuscript.

**Done**

**85** ==> Line 34. I don't think "in situ samplings" is grammatically correct. Please check and modify if necessary.

We have put it in the singular form (see line 38 (page 2) in the revised manuscript).

==> Line 37. "associated to" should be "associated with"

Done, see line 41 (page 2) in the revised manuscript.

90 ==> Line 38. Comma needed after "surface ocean"

Done, see line 42 (page 2) in the revised manuscript.

We have also added a sentence (lines 57-60 (page 3) in the revised manuscript) to highlight the originality of our study, following the comments of the other reviewer.

We have also replaced "low" by "moderately" (lines 56, 66 in page 3 and also in the title), following the comments of the other reviewer.

**Materiels and Methods**

==> If I am not mistaken, you selected the two sampling trajectories based on two regions of Chl-a concentration using the satellite-based SPASSO tool. Based on Figure 1, I can see a region of high Chl-a corresponding to the WE transect, but have trouble distinguishing the second region of unique surface Chl-a that justifies the position of the NS transect. Perhaps it is the colour scale/colorbar limits. Are the regions also selected based on SST and currents? In any case, I would rephrase or try and be more specific of why these two sampling transects were selected, and which areas you are referring to. I believe the colour scale can be improved to highlight this.

105 We have modified this figure, see figure below (i.e., figure 1 (page 24) in the revised

manuscript).

110

(a) Route of the RV Beautemps-Beaupré during PROTEVSMED-SWOT (pink line). The blue box corresponds to the area sampled with Lagrangian strategy. (b) Map of satellite-derived [Chla] provided by CLS for 3 May 2018, selected in the Lagrangian area and superimposed on the route of the ship (black dotted line). The orange and purple lines delimit the two areas called "hippodromes": West-East (orange) and North-South (purple). The red line represents the route of the SeaExplorer glider.

==> I am not really familiar with FSLEs or Langrangian techniques. Thus, out of curiosity, is the FSLE a commonly used index for detecting fronts/fine scale features? Can you provide citations supporting this?

- 115 The first study that showed the interest of using FSLE-derived fronts for biogeochemical studies was probably Lehahn et al., 2007 (see in particular their Fig. 8 and 9). Before, Abraham and Bowen (2002) have been the first to apply the Lyapunov exponent technique (although finite-time, not finite-size) to the ocean, in turn borrowing some ideas from dynamical system theory (see in particular Boffetta et al., 2001). For campaign studies, the
- 120 FSLE analysis permits to identify biogeochemical regions of potential interest. This strategy has already been tested, either in post-cruise or real-time analysis, during many campaigns such as LOHAFEX (Smetacek et al., 2012), Latex10 2010 (Petrenko 2010), KEOPS2 (d'Ovidio et al., 2015), STRASSE 2012, OUTPACE 2015 (Rousselet et al., 2018; de Verneil et al., 2019), OSCAHR 2015 (Marrec et al., 2018; Rousselet et al., 2019), PEACETIME
- 125 2017, SARGASSES 2017, FUMSECK 2019 (Barrillon 2019; Comby et al., 2021), TONGA 2019 (Benavides et al., 2021) and SWINGS 2021 to identify structures of interest. A review on the FSLE and other satellite-based Lagrangian techniques can be found in Lehahn et al., 2018).

In the revised manuscript we have integrated some ideas detailed above, see lines 102-112

**130 (page 4).**

==> Do maps of altimetry/SST also show the existence of the front between the two water masses?

In our work altimetry is the input data for the FLSE (see above). At 38° N 20', the altimetryderived surface current directions change drastically along the NS transect, suggesting a front. However, this is not clearly the case for the WE transect. We don't think altimetric 135 maps are essential for the paper because the front is clearly visible with the VM-ADCP current and the FSLE (see figures above, or eventually in the future supplementary material). Maps of SST can provide another view of fine-scale dynamics. However, the front is not clearly visible on the map of SST. Indeed, gradients of temperature are not enough contrasted in spring, in the Mediterranean Sea. It is easier to locate the front with the map of [Chla] (cf Fig. 1) than the SST, that is why we think these maps aren't necessary for the paper, and we have not added these figures in the revised manuscript.

---

## Referee Report (RR1)

**General comments:**

This is my second time reviewing the manuscript of Tzortiz et al. Overall, the datasets and plots are convincing and the results constitute a novel and interesting contribution to the study of phytoplankton community dynamics in the Mediterranean Sea. I can see that attempts have been made to address my previous comments and indeed, some parts of the manuscript have been improved. However, unfortunately the manuscript appears hastily written at times, and contains sections that are either confusing or grammatically incorrect. In addition, I still feel the authors need to spend some more time on their discussion to really smooth out their interesting story. For examples, the results are extremely detailed, yet the discussion seems speculative and broad at times. Overall, the manuscript has potential, but in my opinion, is currently not at an adequate level for publication in Biogeosciences. I recommend the manuscript for major revisions.

**Specific comments:**

**Abstract:**

Line 2. I would re-phrase "samplings" to something like "in situ – based studies". It sounds a bit strange grammatically otherwise. I would also mention that you are referring to fine scale ocean dynamics when mentioning the lack of in situ data.

Line 8. Please move "at high spatial resolution" to come after "both physical and biogeochemical variables".

Line 9. Particular attention was "*given to*"?

Line 13. I would remove "With respect to previous studies".

**Introduction:**

Line 20. Phytoplankton "**are**"? Please also add "the" before "ocean", and also change ocean to oceans.

Line 21. "It is"? I presume you mean phytoplankton. Please rephrase to "they are" and check the grammar for this elsewhere (e.g., in line 22).

Line 23. Can probably remove Line 23 (e.g., from "Ptacnik…)". Not sure if it really adds anything extra to your first paragraph.

Line 26. Change "but also", to "and also".

Line 57 – 58. Please rephrase this. It sounds a little bit contradictory as you state that the effect of fine scale is non predictable (which in some aspect is what you're trying to do in this manuscript?)…

Line 75. "have allowed **researchers/the authors** to capture"?

Line 82. "dedicated to **providing**"? Would also make "surface current" plural.

**Results**:

Lines 205-213. You state the existence of a fine-scale structure (e.g., line 211) and then state that the FLSE/current direction are *likely* influenced by the presence of a fine scale structure. I would rephrase as this sounds a little bit contradictory.

Line 248. Please correct the grammar in this sentence (e.g., "this can be explained because…"). Also, I'm not sure what you mean by "were realized". Please check carefully your sentences throughout the MS.

Line 252 – 260. Is it possible to just re-arrange this paragraph slightly. I think it would be good to start the paragraph by stating that the two different surface water masses can be clearly seen in the glider transects – your most important result from the plot I suppose? Currently, you focus on distinguishing between surface and intermediate waters.

Some of the figures (e.g., figure 5) are a little difficult to read, particularly in terms of the size of axis labels and the tick markers. If possible, I would increase their size for the reader.

Section 3.3. This is a nice analysis, and the cytometry results are clear. If possible, please try and improve the grammar and smoothness of how the paragraphs are written, as there are several instances where it is difficult to follow (e.g., "the distribution of this latter", "unambiguously put in evidence", "thanks to their light scatter").

**Discussion:**

Lines 368-374.  I can see a deepening of the DCM at around 38 degrees 30', but not sure if you can state that this continues (at least clearly) north of that latitude. The actual maximum Chl-a value (0.8-0.9 µg/l) appears to remain at a fairly constant depth along the transect. That being said, I can see the lower "boundary" of the DCM is generally deeper. I would be careful with the wording here

Line 385. I do not see any correlation analysis presented in Figure A5?

Section 4.2 Although I appreciate the authors considered my previous comment regarding further discussion of this analysis (it was generally skipped over in the last version's discussion), this section is a little bit confusing, especially towards the end. For example, in line 391, the authors state that the high contents of tryptophan and tyrosine *must be* correlated with microphytoplankton. In the next sentence, it states that microphytoplankton *might* produce these fluorophores.  General, loose connections are made and further justification of your statements or a re-wording of the text is needed. Overall, I'm not sure of the strength of the author's discussion here. In addition, you are now discussing the results you have already presented, but rather presenting new results/figures not included in the main manuscript. I don't want to discredit the author's

hard work in this revision, but some improvement is still needed in the flow/organisation of the discussion section.

Section 4.3. Some work is needed here. The last paragraph particularly seems hastily written, is grammatically incorrect and diverges from other parts of the manuscript that are generally well-written. E.g., Lines 430 – 436. It is currently not at publication level.

---

## Referee Report (RR2)

**Tzortzis et al. Review 3**

This is my third time reviewing the manuscript of Tzortzis et al. Overall, I think the authors have done a great job implementing my previous comments and I'm happy to see the manuscript reach the stage it is currently in. I appreciate the efforts of the authors and their attention to improving the grammar of the manuscript. With the implementation of some minor revisions (see below), I believe the manuscript constitutes a useful contribution to Biogeosciences.

Abstract:

Line 9. Would rephrase to "at **a** high spatial resolution"
Line 13. "**Different concentrations of** chlorophyll-a and O2"?
Line 13. Comma after "Here"

Introduction:

Line 31. Remove "**fields and**"?
Line 44. Remove "-well" in "well-known"?
Line 88/89. "**data set**" should be "dataset"

Methods:

Line 120. Comma after "**During the cruise**".

Results:

Line 220. Correct spelling of "**Substancially**" to "Substantially".
Line 247. "**which as**" should be "which has".
Line 303. "**these latter**" to "the latter".

Conclusions:

Line 464. I think you need the word "**Because**".
Line 469. Please correct the spelling of "**recommanded**" to "recommended".
Line 475. Remove "**in**" after "for both".
Line 484-485, I would rephrase to "which will provide a unique opportunity for a more detailed study of physical-biological fine-scale coupling".

---

## Author Response (AR2)

Dear Editor,

We are grateful for your interest in our study, and for accepting our manuscript in revision for a second time. We are also grateful to the two anonymous reviewers for their helpful and constructive comments.

We thank the referee 1 who has decided to accept "as it" our manuscript after the first revision. We also thank referee 2 for his detailed and constructive suggestions that helped us to further improve the presentation of our work. In addition to reworking the scientific aspect we also hired a native English-speaking proofreader to accurately verify the grammar of our text. We hope that our actual revised version, that precisely takes into account all his comments, will be accepted for publication.

Below you find the point-by-point reply to the referee 2 comments.

**Authors's reply to Anonymous Referee #2 comments on bg-2021-38 Tzortzis et al.**

Dear referee,

Thank you very much for the attention that you have given to our work for a second time, as well as for your English corrections. We have reworked the manuscript, taking into account your suggestions. We have also hired a native English-speaking proofreader to accurately verify our manuscript. As in our previous reply, your comments have been copied here in blue, after the $\Rightarrow$ symbol. You can find the revised manuscript and also the marked-up manuscript version showing the changes made on the Biogeosciences website. Note that on the marked-up version the modifications concerning your comments are in red and the English corrections suggested by the English-speaking appear in blue.

**General comments:**

$\Rightarrow$ This is my second time reviewing the manuscript of Tzortiz et al. Overall, the datasets and plots are convincing and the results constitute a novel and interesting contribution to the study of phytoplankton community dynamics in the Mediterranean Sea. I can see that attempts have been made to address my previous comments and indeed, some parts of the manuscript have been improved. However, unfortunately the manuscript appears hastily written at times, and contains sections that are either confusing or grammatically incorrect. In addition, I still feel the authors need to spend some more time on their discussion to really smooth out their interesting story. For examples, the results are extremely detailed, yet the discussion seems speculative and broad at times. Overall, the manuscript has potential, but in my opinion, is currently not at an adequate level for publication in Biogeosciences. I recommend the manuscript for major revisions.

We acknowledge that we were not clear enough in some parts of the manuscript, in particular in sections 4.2 and 4.3 of our discussion. In this revised version of the manuscript, we have improved these sections, taking into account your comments. Furthermore, we have given particular attention to improve English grammar, following your suggestions.

However, we kindly disagree with you on the fact that our discussion is speculative. Indeed, in our discussion we have highlighted important aspects concerning the physical and biological coupling at fine scale, providing an in situ confirmation to the theories advanced by previous numerical simulations studies. And although some points remain uncertain at the end of this very study due to some missing variables (i.e., nutrients), we will improve that in the future thanks to the new campaigns already planned with an optimal sampling strategy.

**Specific comments:**

**Abstract:**

⟹ Line 2. I would re-phrase "samplings" to something like "in situ – based studies". It sounds a bit strange grammatically otherwise. I would also mention that you are referring to fine scale ocean dynamics when mentioning the lack of in situ data.

Following your suggestions, we have modified the abstract (see line 3).

⟹ Line 8. Please move "at high spatial resolution" to come after "both physical and biogeochemical variables".

Following your suggestions, we have modified the abstract (see lines 8).

⟹ Line 9. Particular attention was "given to"?

Following your suggestions, we have modified the abstract (see line 9).

⟹ Line 13. I would remove "With respect to previous studies".

Following your suggestions, we have removed that (see line 13).

**Introduction:**

⟹ Line 20. Phytoplankton "are"? Please also add "the" before "ocean", and also change ocean to oceans.

Following your suggestions, we have corrected that (see line 20).

⟹ Line 21. "It is"? I presume you mean phytoplankton. Please rephrase to "they are" and check the grammar for this elsewhere (e.g., in line 22).

Following your suggestions, we have corrected that (see lines 21-22).

⟹ Line 23. Can probably remove Line 23 (e.g., from "Ptacnik...)". Not sure if it really adds anything extra to your first paragraph.

Following your suggestions, we have removed this sentence.

⟹ Line 26. Change "but also", to "and also".

Following your suggestions, we have modified that (see line 25).

⟹ Line 57 – 58. Please rephrase this. It sounds a little bit contradictory as you state that the effect of fine scale is non predictable (which in some aspect is what you're trying to do in

this manuscript?)...

We have replaced "non predictable" by "more elusive" (Merriam-Webster dictionary: meaning "hard to comprehend or define"),  (see lines 59-60).

⇒ Line 75. "have allowed researchers/the authors to capture"?

Following your suggestions, we have corrected these lines (see line 77).

⇒ Line 82. "dedicated to providing"? Would also make "surface current" plural.

Following your suggestions, we have corrected these lines (see line 83).

**Results:**

⇒ Lines 205-213. You state the existence of a fine-scale structure (e.g., line 211) and then state that the FLSE/current direction are likely influenced by the presence of a fine scale structure. I would rephrase as this sounds a little bit contradictory.

We have removed this line. Indeed, thanks to your comment, we have realized that this sentence was not necessary for the description of our results and that it anticipates our discussion.

⇒ Line 248. Please correct the grammar in this sentence (e.g., "this can be explained because..."). Also, I'm not sure what you mean by "were realized". Please check carefully your sentences throughout the MS.

Following your suggestions, we have modified this part (see lines 256-257).

⇒ Line 252 – 260. Is it possible to just re-arrange this paragraph slightly. I think it would be good to start the paragraph by stating that the two different surface water masses can be clearly seen in the glider transects – your most important result from the plot I suppose? Currently, you focus on distinguishing between surface and intermediate waters.

We are a bit confused as in this paragraph, we are not referring to the glider transects but we focus on the vertical sections of the Seasoar (i.e., Fig. 5 in the paper). However, you are right concerning the structure of this paragraph. Thanks to your suggestions, we have modified the text in the revised manuscript, highlighting our most important result, which is indeed that the two different surface water masses can be clearly identified on these vertical sections. Furthermore, in order to clarify this part, we have moved it above the paragraph concerning the description of the water masses with the $\Theta$-$S_A$ diagrams of the glider (see lines 242-252).

⇒ Some of the figures (e.g., figure 5) are a little difficult to read, particularly in terms of the size of axis labels and the tick markers. If possible, I would increase their size for the reader.

We have increased the size of axis labels and tick markers on figure 5 (see Fig. 5 in the revised manuscript), and also on figure A4 (see Fig. A4 in the revised manuscript).

⇒ Section 3.3. This is a nice analysis, and the cytometry results are clear. If possible, please try and improve the grammar and smoothness of how the paragraphs are written, as there are several instances where it is difficult to follow (e.g., "the distribution of this latter", "unambiguously put in evidence", "thanks to their light scatter").

We have corrected that (see lines 262-264, 267, 285).

**Discussion:**

⇒ Lines 368-374. I can see a deepening of the DCM at around 38 degrees 30', but not sure if you can state that this continues (at least clearly) north of that latitude. The actual maximum Chl-a value (0.8-0.9 µg/l) appears to remain at a fairly constant depth along the transect. That being said, I can see the lower "boundary" of the DCM is generally deeper. I would be careful with the wording here.

In order to follow the evolution of the deepening of the DCM, we have selected the maximum of [Chla] along the outward and the return route of the SeaExplorer glider (see figure below). Indeed, the maximum [Chla] value (0.8-1 µg L$^{-1}$) is constant in depth (it varies between 50 and 80 m) along the outward route of the glider (Fig. a), except at the beginning of the transect where the maximum is located on the surface. On the return route (Fig. b), some of the values are located on the surface, but the majority of the maximum [Chla] value varies between 50 and 80 m as on the outward route. As you pointed out, this is the lower "boundary" of the layer containing the DCM which is deeper. Thanks to your remark, we have modified the text in the revised manuscript (see lines 370-373).

[Figure]

Figure: Vertical profiles of the maximum of [Chla], measured by the SeaExplorer glider, along the outward route (a): 6 May 2018 00:00 - 9 May 2018 21:00 UTC, and along the return route (b): 10 May 2018 00:00 - 13 May 2018 21:00 UTC.

The DCM is clearly identified on the vertical sections of the glider, that is why, we think the figure above is not necessary for the paper, and we have not added this figure in the revised manuscript.

⇒ Line 385. I do not see any correlation analysis presented in Figure A5?

Indeed, this figure represents Box-and-whisker plots and thus does not show any correlation analyses. We apologize for this mistake. However, as mentioned in figure A5 caption, t-test have been performed to compare the mean values of Chla, Peak B and Peak T in "older AW" and in "younger AW". Mean values were significantly higher in "older AW" than in "younger AW" (t-test, $p < 0.0001$). Correlation analyses have been made however on glider data (the two transects from 5 to 200 m depth) to determine correlations between [Chla], [O2], tyrosine-, and tryptophan-like fluorophores. We found a very highly significant linear positive correlation between [Chla] and tyrosine-, and between [O2] and tryptophan-like fluorophores ($r = 0.88$ and $0.84$, $n \sim 32595$, $p < 0.0001$).

Therefore, this sentence has been modified. Please see our answer below since the whole paragraph has been changed (see lines 376-396 in the revised manuscript).

⇒ Section 4.2 Although I appreciate the authors considered my previous comment regarding further discussion of this analysis (it was generally skipped over in the last version's discussion), this section is a little bit confusing, especially towards the end. For example, in line 391, the authors state that the high contents of tryptophan and tyrosine must be correlated

with microphytoplankton. In the next sentence, it states that microphytoplankton might produce these fluorophores. General, loose connections are made and further justification of your statements or a re-wording of the text is needed. Overall, I'm not sure of the strength of the author's discussion here. In addition, you are now discussing the results you have already presented, but rather presenting new results/figures not included in the main manuscript. I don't want to discredit the author's hard work in this revision, but some improvement is still needed in the flow/organisation of the discussion section.

We agree with you that this paragraph deserves clarification and modifications. Also, we propose **to replace the part :**

[revised manuscript text omitted]

See lines 376-396 in the revised manuscript.

⇒ Section 4.3. Some work is needed here. The last paragraph particularly seems hastily written, is grammatically incorrect and diverges from other parts of the manuscript that are generally well-written. E.g., Lines 430 – 436. It is currently not at publication level.

In your general comments, you have highlighted that our discussion seems speculative. As specified in the beginning of this document, we kindly disagree with you on this point. However, we acknowledge that we were not clear enough on this section. Thanks to your comments, we have rearranged this section in order to clarify our development.

We have also given particular attention to improving English grammar.

---

## Author Response (AR3)

Dear Editor,

We are grateful for your interest in our study, and for accepting our manuscript in minor revision. We are also grateful to the two anonymous reviewers for their helpful and constructive comments.
We thank the referee 2 who spent time correcting our manuscript and for his English suggestions that helped us to further improve the presentation of our work. Below you find the point-by-point reply to the referee 2 comments.

Best regards

**Tzortzis et al. Review 3**

This is my third time reviewing the manuscript of Tzortzis et al. Overall, I think the authors have done a great job implementing my previous comments and I'm happy to see the manuscript reach the stage it is currently in. I appreciate the efforts of the authors and their attention to improving the grammar of the manuscript. With the implementation of some minor revisions (see below), I believe the manuscript constitutes a useful contribution to Biogeosciences.

⇒ Thank you very much for the attention that you have given to our work for a third time, as well as for your English corrections. We have reworked the manuscript, taking into account your suggestions.

**Abstract:**

Line 9. Would rephrase to "at a high spatial resolution"

⇒ Following your suggestions, we have rephrased (see line 8).

Line 13. "Different concentrations of chlorophyll-a and O2"?

⇒ Following your suggestions, we have modified that (see line 13).

Line 13. Comma after "Here"

⇒ We have added a comma (see line 13).

**Introduction:**

Line 31. Remove "fields and"?

⇒ We have removed that (see line 31).

Line 44. Remove "-well" in "well-known"?

⇒ We have corrected that (see line 43).

Line 88/89. "data set" should be "dataset"

⇒ We have modified that (see line 86).

**Methods:**

Line 120. Comma after "During the cruise".

⇒ We have added a comma (see line 115).

**Results:**

Line 220. Correct spelling of "Substancially" to "Substantially".

⇒ Thank you, we have corrected (see line 210).

Line 247. "which as" should be "which has".

⇒ We have corrected (see line 236).

Line 303. "these latter" to "the latter".

⇒ We have modified that (see line 288).

**Conclusions:**

Line 464. I think you need the word "Because".

⇒ We have added this word (see line 447).

Line 469. Please correct the spelling of "recommanded" to "recommended".

⇒ Thank you, we have corrected (see line 452).

Line 475. Remove "in" after "for both".

⇒ We have removed "in" (see line 457).

Line 484-485, I would rephrase to "which will provide a unique opportunity for a more detailed study of physical-biological fine-scale coupling".

⇒ Thank you, we have rephrased following your suggestion (see line 466).